# A comprehensive characterization of the cell-free transcriptome reveals tissue- and subtype-specific biomarkers for cancer detection

Matthew H. Larson [1,2 ✉], Wenying Pan[1,2], Hyunsung John Kim[1], Ruth E. Mauntz[1], Sarah M. Stuart[1], Monica Pimentel[1], Yiqi Zhou[1], Per Knudsgaard [1], Vasiliki Demas[1], Alexander M. Aravanis[1] & Arash Jamshidi[1]

Cell-free RNA (cfRNA) is a promising analyte for cancer detection. However, a comprehensive assessment of cfRNA in individuals with and without cancer has not been conducted. We perform the first transcriptome-wide characterization of cfRNA in cancer (stage III breast [$n = 46$], lung [$n = 30$]) and non-cancer ($n = 89$) participants from the Circulating Cell-free Genome Atlas (NCT02889978). Of 57,820 annotated genes, 39,564 (68%) are not detected in cfRNA from non-cancer individuals. Within these low-noise regions, we identify tissue- and cancer-specific genes, defined as "dark channel biomarker" (DCB) genes, that are recurrently detected in individuals with cancer. DCB levels in plasma correlate with tumor shedding rate and RNA expression in matched tissue, suggesting that DCBs with high expression in tumor tissue could enhance cancer detection in patients with low levels of circulating tumor DNA. Overall, cfRNA provides a unique opportunity to detect cancer, predict the tumor tissue of origin, and determine the cancer subtype.

[1] GRAIL, Inc., Menlo Park, CA, USA. [2]These authors contributed equally: Matthew H. Larson, Wenying Pan. ✉email: mlarson@grailbio.com

Tumor-derived cell-free DNA (cfDNA) has emerged as an effective biomarker for cancer detection. The rapid reduction in sequencing costs, combined with more efficient library preparation techniques, has enabled the detection of cancer-associated point mutations, copy number variations, and methylation markers at increasingly earlier stages of disease[1–4]. Despite the promise of these methods for cancer screening, they are fundamentally constrained by the amount of tumor DNA shed into the blood during cell death[5]. Small or slow-growing tumors release less DNA into circulation[6], leading to low tumor fractions and reduced sensitivity for early-stage cancer detection. Furthermore, most cfDNA features (e.g., small nucleotide variants) are not tissue-specific, making it difficult to predict the tumor tissue of origin (TOO) in positively screened patients with cancer. More recently, targeted analysis of methylation markers on cfDNA has been demonstrated to detect and localize cancer with high specificity[7]. However, detection and localization of cancer at its earliest stages will benefit from the exploration of additional biomarkers to complement detection by cfDNA.

Previous reports suggest that cancer cells also release cell-free RNA (cfRNA) into circulation[8–12]. Cell-free RNA presents an opportunity to detect cancer in patients with low tumor shedding rates, as overexpression of tumor-specific transcripts could lead to amplification of tumor-derived RNA signals in the blood. In addition, cfRNA may be released into the blood through mechanisms other than cell death, such as exosome-mediated signaling by living cells[13]. Consequently, tumor-derived cfDNA and cfRNA may originate from distinct cell populations within the tumor microenvironment, potentially expanding the opportunities for cancer detection through the combined screening of multiple analytes in the blood.

Historically, the study of cfRNA has focused on either microRNAs (miRNA) or a small number of known cancer-related messenger RNAs (mRNA). miRNAs are stable and relatively abundant in plasma[14,15], and numerous studies have demonstrated cancer-associated miRNA expression changes[16]. However, miRNA levels can be skewed by preanalytical processing conditions, quantification strategies, and batch effects, which have translated to a lack of reproducibility, low interpretability, and poor specificity for miRNA biomarkers[17]. Tumor-derived cell-free mRNA was identified nearly 2 decades ago by polymerase chain reaction (PCR)[9], but studies of cell-free mRNA are typically hypothesis-driven, focusing on mutations or expression changes for previously characterized oncogenes[9,10,12]. As such, these studies may miss a large number of potential biomarkers. Furthermore, many cancer-related transcripts are also highly abundant in circulating immune cells, red blood cells, and platelets, making it difficult to identify small expression changes associated with early-stage disease.

In this study, we aim to characterize cfRNA and to identify cell-free biomarkers specific to patients with breast and lung cancer. We develop a strategy to preserve, extract, and sequence extracellular mRNAs from patient plasma while avoiding confounding background noise from circulating blood cells, and use this approach to conduct a transcriptome-wide characterization of circulating RNA in the blood of cancer and non-cancer participants from the Circulating Cell-free Genome Atlas Study (CCGA, NCT02889978). Here, we define the baseline cell-free transcriptome in the absence of cancer and identify tissue and subtype-specific cfRNA biomarkers in breast and lung cancer patients. Taken together, these results suggest an opportunity to detect and localize cancer using cell-free mRNA and provide a framework for identifying highly specific biomarkers in different cancers.

## Results

**Analytical characterization of cell-free RNA.** We systematically evaluated the effect of preanalytical parameters on cfRNA yield and expression profiles to establish a reproducible protocol for preserving cfRNA in patient plasma (see Supplementary Materials regarding the effect of blood collection tube type [Supplementary Figs. 1–3], shipping temperature [Supplementary Fig. 4], plasma storage [Supplementary Fig. 5], and freeze/thaw cycles [Supplementary Fig. 6] on cfRNA recovery). Based on these studies, we determined that Streck Cell-free DNA blood collection tubes preserve cfRNA in whole blood at temperatures ranging from 6–35 °C for up to 48 h prior to plasma separation. Moreover, we found that cfRNA is stable in frozen plasma for up to 12 months when collected in Streck tubes and stored at −80 °C and that a single freeze/thaw cycle does not significantly affect cfRNA yield, enabling the use of banked plasma for the study of cfRNA. We selected stored plasma from 47 breast and 32 lung cancer subjects (stage III) collected as part of the CCGA study (see Materials and Methods for sample selection criteria). Ninety-three frequency age-matched individuals without cancer were also selected to characterize the baseline transcriptome in non-cancer individuals and ensure high specificity of the identified RNA biomarkers (a full list of study participants can be found in Supplementary Table 5).

Extracted and deoxyribonuclease (DNase)-treated cfRNA contained a large proportion (55 ± 12%, median ± standard deviation [SD]) of small RNA fragments (<200 nucleotides [nt]), consistent with the previous reports[14]. We also observed longer fragments (200–2000 nt) and prominent 18S/28S ribosomal RNA peaks (Fig. 1a), suggesting that a significant proportion of circulating RNA is protected from RNase degradation in the blood and remains largely unfragmented. The average cfRNA yield from non-cancer and cancer samples was 7.1 and 7.9 ng/mL of plasma, respectively, which is comparable to cfDNA yields from the same donors (9.9 and 8.9 ng/mL of plasma in the non-cancer and cancer groups) (Supplementary Fig. 7). We also observed substantial patient-to-patient variability in cfRNA yield (83% and 81% coefficient of variation in the non-cancer and cancer groups, respectively).

Whole-transcriptome RNA-seq libraries were prepared for each patient using the entire yield of extracted cfRNA (see Methods and Supplementary Fig. 8). The sequencing results for these whole-transcriptome libraries showed that cfRNA was dominated by ribosomal RNA (rRNA) and mitochondrial rRNA (Mt rRNA), which accounted for >95% of sequenced transcripts (Fig. 1a), while mRNA made up a relatively small fraction of cfRNA (~2%), consistent with the relative proportions of rRNA-to-mRNA in most cell types. Within the population of cell-free mRNAs, a few high-abundance transcripts accounted for half of all sequenced transcripts. Among these transcripts were blood-derived mRNAs such as the hemoglobin (HB) subunits (*HBA1*, *HBA2*, *HBB*, *HBD*); components of the major histocompatibility complex (*B2M*, *CD74*); and the RNA component of the signal recognition particle (*RN7SL*), which has been identified as a membrane-associated transcript present at high levels in red blood cells[18]. We depleted these high-abundance transcripts after amplification of complementary DNA (cDNA) libraries to minimize losses at the beginning of our library preparation protocol when the material is limiting. After depletion, the resulting RNA-seq libraries were sequenced to saturation and analyzed using a custom bioinformatics pipeline that generated unique molecular identifier (UMI)-collapsed counts for each gene on a sample-by-sample basis. Compared to cfDNA, which shows relatively even coverage across the genome due to the release of 2 genomic copies during cell death, cfRNA shows increased coverage in exonic regions of expressed genes (Fig. 1b),

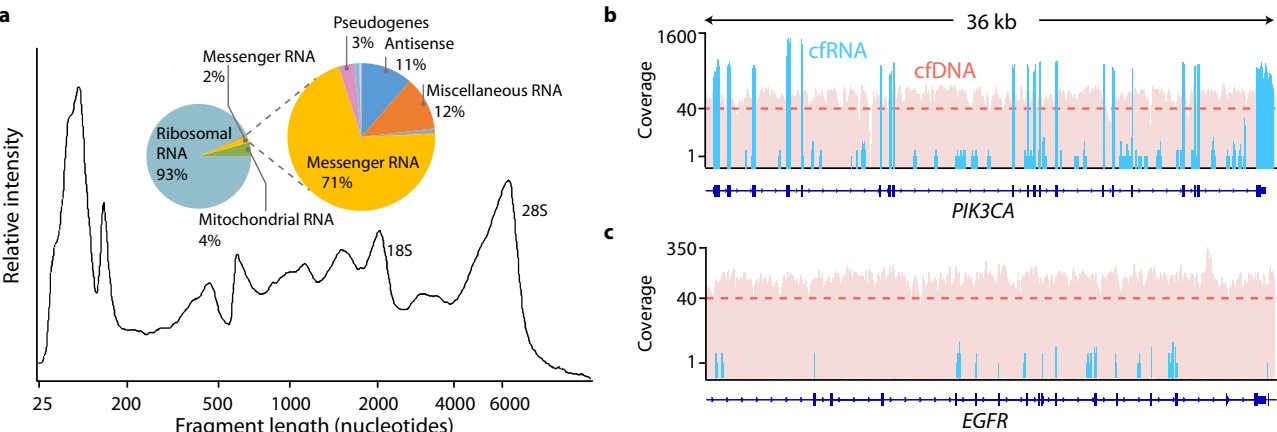

**Fig. 1 Analytical characterization of cell-free RNA. a** Fragment Analyzer (Agilent) trace of cfRNA fragment lengths in a non-cancer sample following deoxyribonuclease (DNase) digestion. Inset: Relative proportion of different RNA types found by whole-transcriptome sequencing in a representative non-cancer sample prior to abundant transcript depletion. Sequencing coverage across a 36 kb region of **b** *PIK3CA*, a high-abundance cell-free RNA gene, and **c** *EGFR*, a low-abundance cell-free RNA gene, from a representative patient sample. Blue bars represent coverage for a whole-transcriptome cfRNA sample with 584 M paired-end reads, and red bars represent coverage for a whole-genome cfDNA sample from the same patient with 871 M paired-end reads. The dashed line represents median cfDNA coverage.

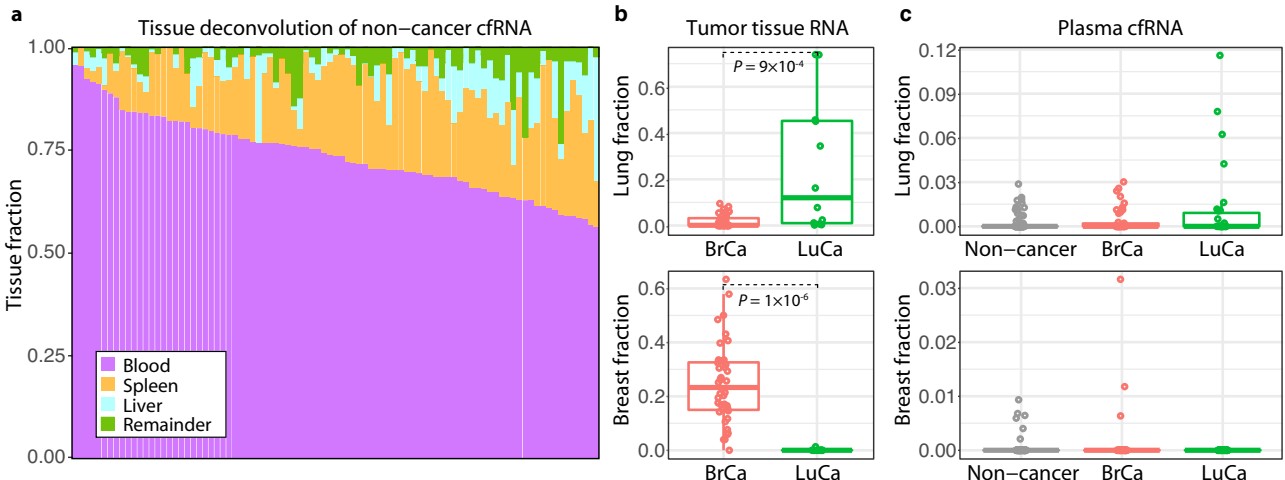

**Fig. 2 Tissue deconvolution of RNA from plasma and tissue samples. a** Tissue deconvolution for cell-free RNA (cfRNA) from 89 non-cancer plasma participants. Each stacked bar represents a single participant. **b** Lung and breast fraction distribution in RNA from matched tumor tissue obtained from 40 breast cancer (BrCa) and 11 lung cancer (LuCa) patients. *P* values from the two-sided Wilcoxon rank-sum method indicate significance levels for differences in tissue-specific fractions across sample groups. **c** Lung and breast fraction distribution in plasma-derived cfRNA from different sample groups (46 BrCa, 28 LuCa, 89 non-cancer). Boxplots indicate the 25% (lower hinge), 50% (horizontal line), and 75% quantiles (upper hinge) of the tissue fraction distribution, with whiskers that indicate observations outside the hinge ± 1.5 x interquartile range (IQR). Outliers (beyond 1.5 × IQR) are plotted individually.

as expected for mature mRNA, and reduced coverage in unexpressed genes (Fig. 1c), reflecting contributions from cell types with different expression profiles.

**Identification of cell-free mRNA biomarkers.** To identify the sources of RNA normally present in circulation, we performed a tissue deconvolution of cfRNA from non-cancer samples (see "Methods" section). The majority of circulating transcripts (74 ± 10%, median ± SD) are derived from blood cells (Fig. 2a), reflecting the high abundance of these cell types in circulation and their constant turnover. We also observed significant contributions from transcripts expressed in the liver and spleen, both of which are involved in blood-filtering and have direct contact with the circulatory system. This is similar to cfDNA origins in plasma, which are also dominated by contributions from white blood cells and liver tissue[19], suggesting that cfRNA and cfDNA in non-

cancer plasma share a common etiology derived from normal turnover of blood cells and blood-filtering organs.

We also performed a tissue deconvolution analysis on RNA from the cancer group, using plasma and matched tumor tissue samples obtained for the CCGA study. The results of these analyses revealed larger contributions from breast and lung tissue in RNA from tumor tissue samples (Fig. 2b), as expected. Notably, lung and breast-derived transcripts were also observed in cfRNA from cancer plasma samples (Fig. 2c). The relative fraction of these tissue-specific transcripts observed in plasma is low compared to estimates in tumor tissue, as expected based on previous estimates of circulating tumor DNA in plasma[6], but are indicative of tissue-specific transcripts in cancer plasma that can be differentiated from the high background of blood-cell derived transcripts. Overall, this suggests that there is a detectable population of tumor-derived RNA circulating in cancer plasma.

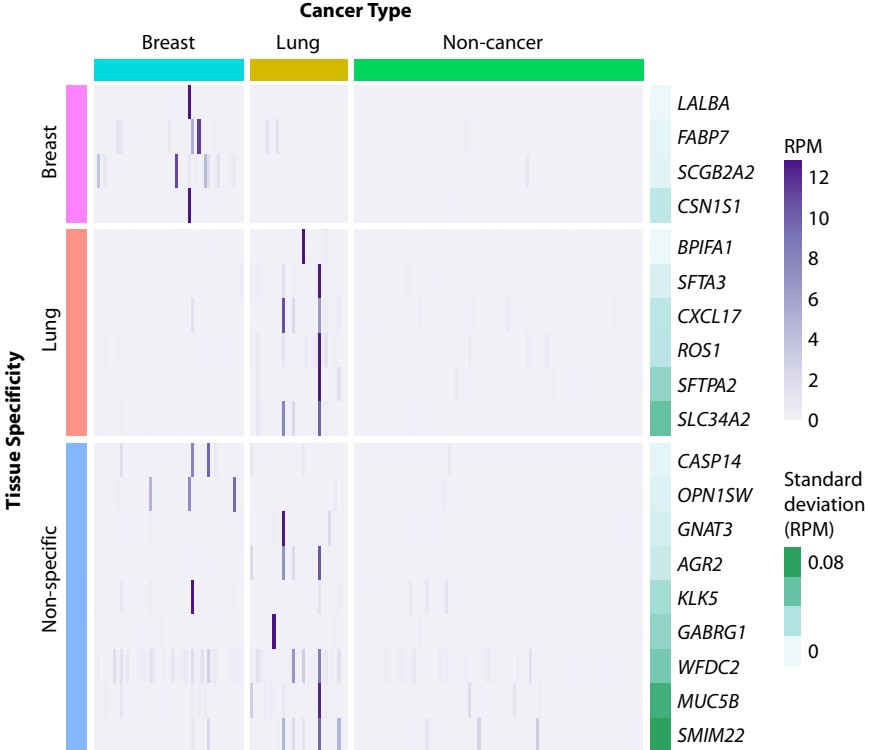

**Fig. 3 Dark channel biomarker (DCB) expression in cell-free RNA from breast, lung, and non-cancer plasma samples.** Samples are shown in columns and DCB genes in rows. Cancer type is indicated above the heatmap, and the tissue specificity of each DCB, as annotated in the Human Protein Atlas (version 18.1), is indicated on the left side of the heatmap. Gene expression values in reads per million (RPM) are represented by the purple gradient and are scaled for visualization purposes. Darkness in non-cancer plasma samples is illustrated by the standard deviation of RPM (green gradient bar) as shown on the right side of the heatmap.

We set out to identify tumor-specific transcripts among the background of RNA from healthy blood cells by focusing our search on genes rarely detected in the plasma of non-cancer individuals. Of 57,820 annotated genes in the GENCODE v19 reference, 39,564 (68%) were absent in non-cancer plasma (median reads per million [RPM] = 0). We call these "dark channels," as they represent regions of the genome that are free of background expression in circulating blood cells. Remarkably, we identified a subset of genes recurrently detected in patients with cancer within these dark channel regions. These genes met the following criteria: (1) they were not detected in non-cancer plasma, (2) they were upregulated in the cancer group compared to the non-cancer group, and (3) they were detected in more than one cancer sample in our cohort (to exclude single outliers). Because these biomarkers are found in regions that are normally "dark" in the non-cancer group, we refer to them as dark channel biomarker (DCB) genes. Overall, 12 DCB genes were identified in lung cancer samples (*SLC34A2*, *GABRG1*, *ROS1*, *AGR2*, *GNAT3*, *SFTPA2*, *MUC5B*, *SFTA3*, *SMIM22*, *CXCL17*, *BPIFA1*, *WFDC2*), and 8 DCB genes were identified in breast cancer samples (*CSN1S1*, *FABP7*, *OPN1SW*, *SCGB2A2*, *LALBA*, *CASP14*, *KLK5*, *WFDC2*) (Fig. 3). One DCB gene (*WFDC2*) was identified in both breast and lung cancer samples.

**Dark channel biomarkers are tissue-specific and subtype-specific.** Unexpectedly, DCB genes are highly enriched for tissue-specific genes. Among all 57,820 annotated human genes in the GENCODE v19 reference, only 0.3% are lung-specific and 0.2% are breast-specific (as defined by the Human Protein Atlas database, version 18.1)[20]. By comparison, 50% of the lung DCB genes (6 of 12) are lung-specific, and 50% of the breast DCB genes (4 of 8) are breast-specific, indicating significant enrichment of

tissue-specific markers for both lung (Fisher's exact P value = $7 \times 10^{-13}$) and breast DCBs (Fisher's exact test P value = $3 \times 10^{-9}$). Furthermore, the tissue specificity of DCBs matched the cancer type of the patients in which they were found: breast-specific and lung-specific DCBs were detected in the plasma of patients with breast and lung cancer, respectively, suggesting an opportunity to predict tumor TOO using cfRNA (Fig. 3).

In addition to their specificity to the tumor TOO, ~30% of the DCB genes identified are also specific to a certain cancer subtype. In our breast cohort, *FABP7* was upregulated in cfRNA from patients with triple-negative breast cancer (TNBC) but downregulated in hormone receptor-positive (HR+) breast cancer (Fig. 4a). Conversely, the DCB gene *SCGB2A2* was downregulated in TNBC but upregulated in HR+ breast cancer samples (Fig. 4a). Analysis of matched tumor tissue expression from 33 breast subjects (23 HR+, 10 TNBC) collected at the time of draw also showed similar results: *FABP7* was upregulated in the tumor tissue of patients with TNBC breast cancer, while *SCGB2A2* was upregulated in tumor tissue from patients with HR+ breast cancer (Fig. 4b). This concordance suggests that the subtype specificity of these DCB genes in plasma reflects their subtype-specific expression in tumor tissue. An analysis of DCB expression in breast cancer tumor samples from The Cancer Genome Atlas (TCGA) also confirms the subtype-specific expression of these markers in a larger cohort of tumor tissue samples (Fig. 4c).

Similarly, this DCB subtype specificity also extends to lung cancer: 4 of the 6 lung DCB genes (*CXCL17*, *SFTA3*, *SFTPA2*, and *SLC34A2*) were upregulated in the plasma of patients with lung adenocarcinoma compared to those with squamous cell carcinoma (Fig. 4d). This same expression pattern was also observed in 7 matched lung cancer tissue samples (4 squamous cell

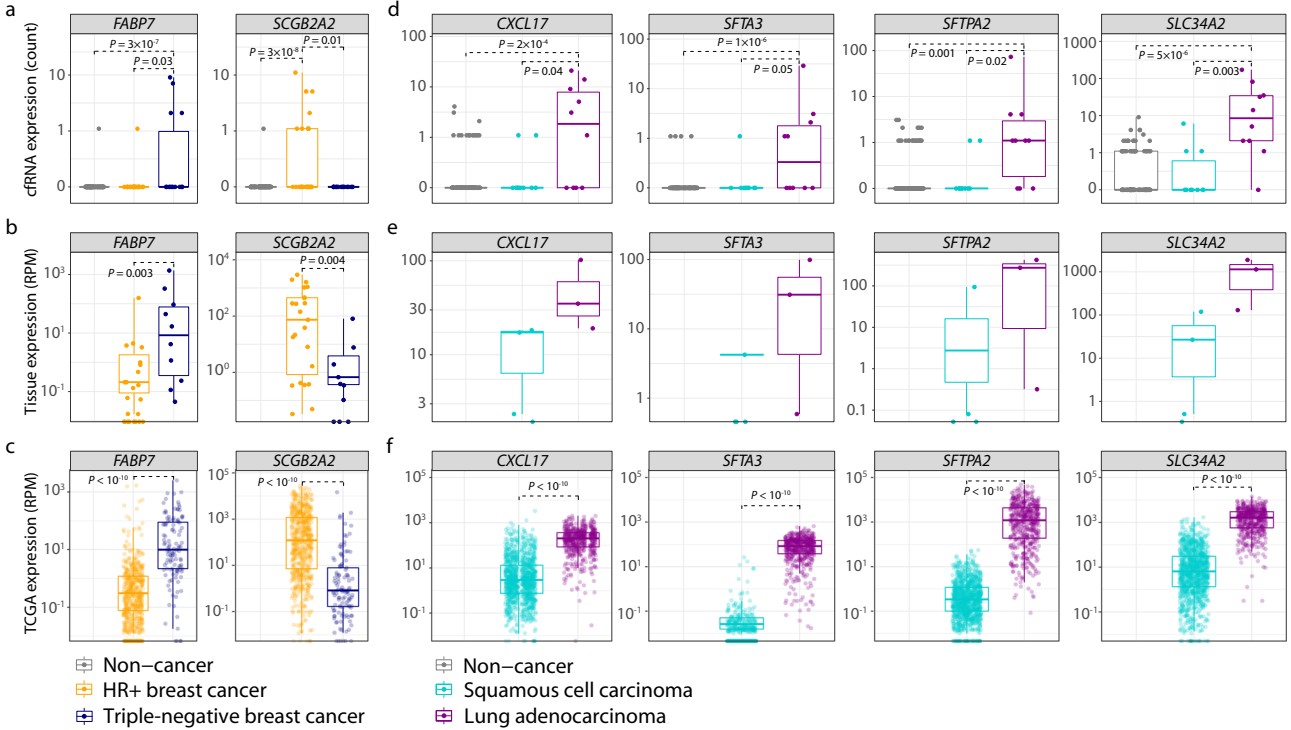

**Fig. 4 Dark channel biomarker (DCB) genes exhibit cancer and subtype-specific expression in cfRNA and tumor tissue.** Expression (strict counts) of breast cancer-specific DCBs in **a** cfRNA of breast cancer ($n = 24$ HR+, $n = 14$ triple negative) and non-cancer participants ($n = 89$), and in **b** matched breast tumor biopsies ($n = 23$ HR+, $n = 10$ triple-negative). **c** Expression (RPM) of RNA in breast tumor tissue ($n = 575$ HR+, $n = 115$ triple-negative) from The Cancer Genome Atlas (TCGA). Expression (strict counts) of lung cancer-specific DCBs in **d** cfRNA of lung cancer ($n = 10$ adenocarcinoma, 10 squamous cell carcinoma) and non-cancer participants ($n = 89$), and in **e** matched lung tumor biopsies ($n = 4$ squamous cell carcinoma, $n = 3$ adenocarcinoma). **f** Expression (RPM) of RNA in lung tumor tissue ($n = 1102$ squamous cell carcinoma, $n = 533$ adenocarcinoma) from TCGA. $P$ values from the two-sided Wilcoxon rank-sum method indicate significance levels for differential expression between cancer subtypes. Boxplots indicate the 25% (lower hinge), 50% (horizontal line), and 75% quantiles (upper hinge), with whiskers that indicate observations outside the hinge ± 1.5 × interquartile range (IQR). Outliers (beyond 1.5 × IQR) are plotted individually.

carcinoma, 3 adenocarcinoma) and tumor tissue samples from TCGA (Fig. 4e, f), highlighting the concordance between lung DCB levels in plasma and tumor tissue. The remaining 2 lung-specific DCB genes (*ROS1*, *BPIFA1*) were also increased in the plasma of patients with lung adenocarcinoma, though it did not rise to the level of statistical significance. However, an analysis of these markers in TCGA tissue data indicates a statistically significant increase in expression in adenocarcinoma tumor tissue compared to squamous cell carcinoma (Supplementary Fig. 9), suggesting that larger cohorts of patients with lung cancer and cfRNA expression data may confirm the subtype specificity of these DCB genes.

**Detectability of DCB genes is correlated with tumor fraction in plasma and expression in tumor tissue.** We observed that some plasma samples in the cancer group did not have detectable levels of DCB genes and set out to determine what governs DCB detectability in cfRNA. Previous studies of cfDNA indicate that the fraction of tumor-derived material in circulation varies among cancers and individuals. If the fraction of tumor-derived cfDNA and cfRNA are proportional for a given patient, then the detectability of a DCB gene in cfRNA should correlate with tumor fraction estimates from cfDNA. As part of the CCGA study, tumor fraction estimates were calculated for all patients with matched tissue using targeted DNA sequencing (Supplementary Materials)[7]. Due to the relatively large number of patients with breast cancer with matched tissue and tumor fraction estimates in our cohort ($n = 33$), we applied this

analysis to breast biomarkers *FABP7* and *SCGB2A2*, which were selected from 4 breast-specific DCBs due to their high recurrence rate among patients with breast cancer (13% and 24%, respectively) and specificity to different breast cancer subtypes. We observed that all 4 patients with breast cancer who had *FABP7* detected in plasma had relatively high tumor fractions (>1%) (Fig. 5a), consistent with the hypothesis that tumor shedding rate is correlated with DCB detection. However, the 5 breast cancer patients with the highest apparent tumor fraction did not have detectable levels of *FABP7* in plasma, suggesting that other factors beyond tumor fraction may affect DCB abundance in cfRNA.

We hypothesized that the detectability of a DCB gene in cfRNA is further modulated by the expression of DCB genes in the matched tumor tissue, with a higher expression in tumor cells leading to an increase in the likelihood that the transcript would be shed and detected in plasma. Consistent with this hypothesis, we observed that DCB gene levels in plasma are correlated with DCB expression in tumor tissue (Supplementary Fig. 10). Motivated by the observation that neither tumor fraction nor expression fully explained the prevalence of DCBs in plasma, we derived a variable—"tumor content"—which is the product of tumor fraction and expression of the DCB gene in matched tumor tissue. Tumor content was more significantly associated with DCB detection in plasma compared to either tumor fraction or tissue expression alone for the detectability of *FABP7* (two-sided Mann–Whitney *U* test *P* value 0.003 vs. 0.02 and 0.01, respectively) and *SCGB2A2* (*P* value 0.0001 vs. 0.01 and

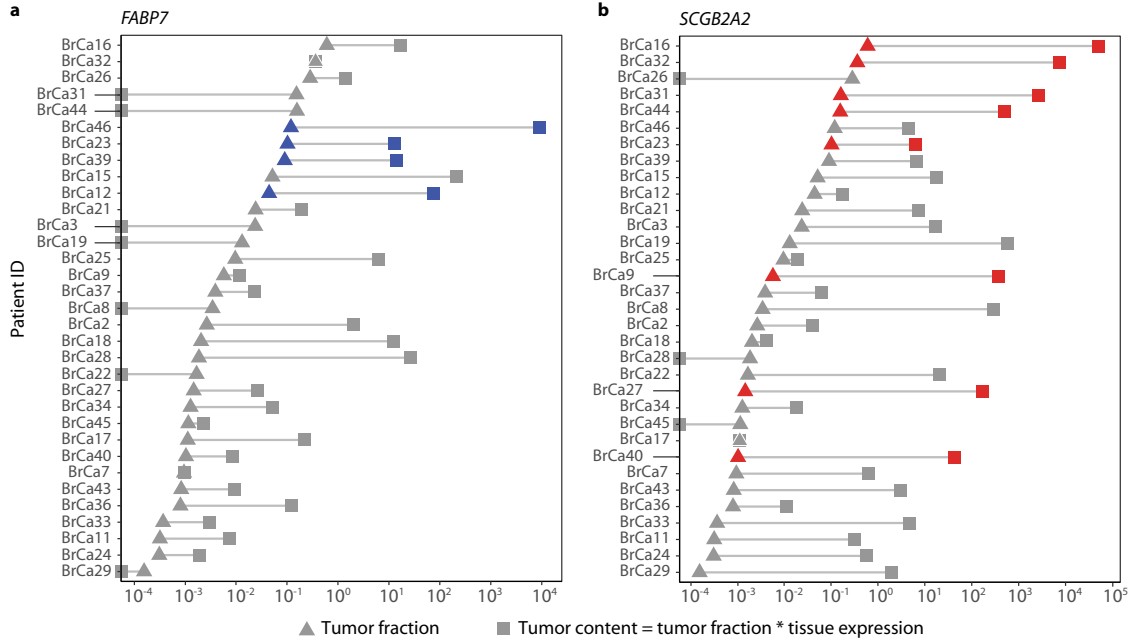

**Fig. 5 The impact of tumor fraction and tumor content on the detectability of dark channel biomarker genes in cell-free RNA.** Patient IDs plotted as a function of tumor fraction (triangles) and tumor content (squares) for **a** *FABP7* and **b** *SCGB2A2*. Patient IDs are ranked in descending order based on patient-specific tumor fractions and are ordered identically in both panels. Blue and red symbols represent samples for which either *FABP7* or *SCGB2A2* was detected in plasma, respectively.

0.001, respectively) (Fig. 5). This may explain why patients with high tumor fraction (>1%) but no expression of *FABP7* in matched tissue did not show detectable levels of *FABP7* in plasma (see samples BrCa31, BrCa44, BrCa3, and BrCa19 in Fig. 5a) and why patients with lower tumor fractions but high tumor tissue expression of *SCGB2A2* showed detectable levels of *SCGB2A2* in plasma (see samples BrCa9, BrCa27, and BrCa40 in Fig. 5b). There are also a number of samples that appear to have relatively high tumor content (>1%) for SCGB2A2 and FABP7, but for which the genes are not detected in cfRNA, suggesting that other factors beyond tumor tissue expression may influence the detectability of a given biomarker in circulation.

Inspired by the observation that tumor content for a given gene is a better predictor of its expression in cfRNA, we developed a statistical method called heterogeneous differential expression analysis (heteroDE) to identify tumor-derived cfRNA biomarkers (see "Methods" section). Standard differential expression techniques, such as DEseq2 and edgeR[21,22] separate samples into 2 categorical groups (disease and control) and assume that samples in each group share the same mean expression value. However, in highly heterogeneous samples like cfRNA, the mean expression of a gene can vary significantly from sample to sample based on the subtype of the primary tumor and the patient-specific tumor fraction. Thus, heteroDE uses the continuous covariate tumor content rather than the categorical covariate group label in the generalized linear model. When we applied heteroDE to breast cancer samples, we identified 7 cfRNA biomarkers (*SCGB2A2*, *CASP14*, *FABP7*, *CRABP2*, *VGLL1*, *SERPINB5*, *TFF1*), 3 of which (*FABP7*, *SCGB2A2*, *CASP14*) overlap with previously identified DCB genes. In summary, we identified a total of 23 cfRNA biomarkers in breast and lung cancer (Supplementary Table 6) using 2 different feature selection methods (DCB criteria and heteroDE). The overlap of genes identified by these two methods suggests that the approaches are complementary and can be used in different circumstances based on the availability of matched tumor tissue.

**Validation of cfRNA biomarkers in a separate cohort**. We set out to validate the 23 cell-free mRNA biomarkers identified in our CCGA cohort in an orthogonal set of breast (*n* = 38) and lung (*n* = 18) cancer plasma samples obtained from a commercial vendor (Discovery Life Sciences). Stage I–IV patients were selected to assess the prevalence of RNA biomarkers across disease progression, and 32 age-matched non-cancer samples were included as controls of expression in patients without cancer. Most of the cancer patients in this cohort do not have reliable subtype data available at the time of collection, so this cohort could only be used to validate the cancer specificity of DCB genes and not their subtype specificity. To improve sensitivity and reduce sequencing requirements, we developed a targeted enrichment approach to select 23 cell-free mRNA biomarkers (Fig. 6a). We also enriched for 34 positive control genes that are normally present in non-cancer plasma (Supplementary Table 7), which act as carrier material in the enrichment step. When compared to the whole-transcriptome assay, we found that the targeted approach increased conversion efficiency for targeted cfRNA transcripts by 2-fold (Supplementary Fig. 11).

Consistent with our previous findings, and despite the increased efficiency of the targeted assay, all but 1 of 23 DCB genes tested in our validation cohort were dark (median RPM = 0) in the non-cancer group (Fig. 6a, Supplementary Table 6), satisfying the expression level criteria for dark channels. Of the 21 genes in our panel detected in >1 cancer sample in the validation cohort, 15 genes were differentially expressed in at least 1 cancer group. Seven DCB genes were differentially expressed in breast cancer (Fig. 6b) and 14 were differentially expressed in lung cancer with a false discovery rate ≤1% (Fig. 6c). The level of cfRNA biomarker expression in the cancer group generally increased with stage, with the highest expression seen for stage IV samples (Fig. 6a and Supplementary Fig. 12). In summary, these results confirm the diagnostic potential of 15 of the 23 DCB genes in individuals with cancer and validate the DCB discovery approach.

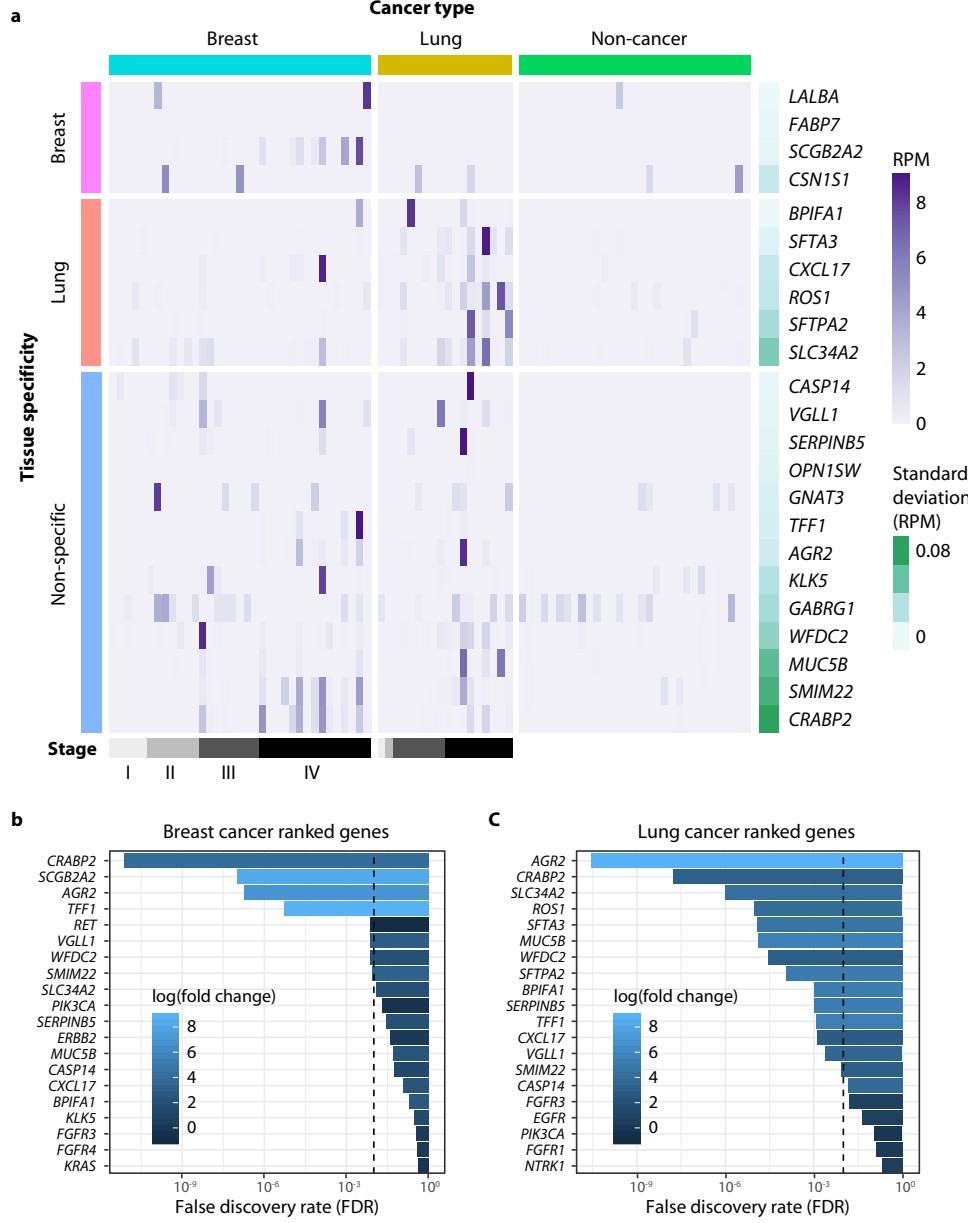

**Fig. 6 Detection of dark channel biomarker (DCB) genes in cell-free RNA of an independent validation cohort. a** Samples are shown in columns and DCB genes in rows. Cancer type is indicated above the heatmap, and tissue specificity of each DCB, as annotated in the Human Protein Atlas (version 18.1), is indicated on the left side of the heatmap. Gene expression values in reads per million (RPM) are represented by the purple gradient and are scaled for visualization purposes. Darkness in non-cancer plasma samples is illustrated by the standard deviation of RPM (green gradient bar) as shown on the right side of the heatmap. False discovery rates (FDR) for the top 20 genes ranked by differential expression analysis of the targeted assay are shown for **b** breast cancers and **c** lung cancers. FDR was calculated as the Benjamini–Hochberg corrected *P* value (using a cut-off of 1% to allow for <1 false-positive across 35 genes).

## Discussion

Early diagnosis of cancer can dramatically improve the chances of survival[23], and the detection of solid tumors through the analysis of circulating nucleic acids in liquid biopsies has demonstrated that cancers can be detected at increasingly earlier stages of disease[7]. As a result, a significant amount of effort has been devoted to the study of cfDNA and its potential for cancer screening. By comparison, relatively little is known about the biology of cfRNA, despite its potential to non-invasively report on transcription dynamics within the tumor environment.

To our knowledge, our study represents the first transcriptome-wide assessment of cfRNA in cancer and non-cancer patients.

Previous work largely focused on circulating miRNA[16], which can be difficult to interpret due to their pleiotropic effects on gene regulation, and which have rarely translated into clinical practice[17]. Studies of circulating mRNA have more clinical applications but are limited to the detection of known oncogenic mRNA markers, often with prior knowledge of mutations harbored in matched tumor tissue[24,25]. Such approaches are invaluable for treatment guidance and monitoring but do not address the potential application of cfRNA for cancer screening. Our results shed light on the origins of cfRNA in plasma and highlight a population of circulating mRNAs that are specific to breast and lung cancer. These results have significant implications for the potential use of cfRNA in cancer detection and monitoring and

provide a framework for identifying highly specific cfRNA biomarkers in other cancers.

The existence of RNA in plasma has been well-established for more than 20 years. However, several factors may have hindered its widespread adoption for cancer screening. First, circulating RNA is assumed to be unstable and highly fragmented. This perception arises from the relative instability of RNA compared to DNA, which itself is highly fragmented and unstable in the blood[26], as well as the high concentration of RNases present in circulation[27,28]. By evaluating different preanalytical conditions and optimizing our extraction protocol, we were able to define a set of conditions for which cfRNA is preserved in plasma after blood collection. By design, our approach to cfRNA extraction is agnostic to the origin of the extracellular material and seeks to isolate all available RNA from the cell-free fraction. This precludes speculation about the potential function of these transcripts in circulation but allows a comprehensive characterization of cfRNA. We recovered full-length mRNAs that are similar to the profiles reported in the literature for exosomal RNA[29], and the total yield of extracted cfRNA is comparable to cfDNA in matched patients. These results suggest that cfRNA is relatively stable in plasma and provide a preanalytical workflow for future studies seeking to quantify circulating RNA in a robust and reproducible manner.

A primary innovation in this work is that we restricted our analysis to genes that are never (or rarely) detected in the non-cancer group, and identified biologically-relevant transcripts that are detected specifically in cancer plasma. We call these genes "dark channel biomarkers." These DCBs address a major challenge: identification of reliable RNA biomarkers for cancer detection while avoiding a large number of false positives. By focusing our analysis on regions that are free of background signals from healthy cells, we can ensure high cancer specificity of the identified biomarkers. This approach effectively reduces the likelihood of identifying false positive cfRNA signal in plasma (e.g., due to technical variations or batch effects) that may arise from analyses focused on fold-changes in expression between cases and controls. We also developed a differential expression analysis framework (heteroDE) that utilizes gene expression from matched breast tumor tissue and tumor fraction from matched cfDNA libraries to identify biomarkers across heterogeneous samples where both tumor fraction and tumor tissue expression can vary dramatically. Using these 2 approaches, we were able to identify 23 cfRNA biomarkers that are robust to processing conditions and specific to cancer (Supplementary Table 6).

Dark channel biomarker genes exhibit several distinct characteristics that support their validity as cancer-specific biomarkers: (1) DCB genes are highly enriched for tissue-specific genes, (2) their expression is correlated with expression in matched tumor tissue, and (3) they have been previously identified as subtype-specific biomarkers in tumor tissue.

SCGB2A2 (mammaglobin-A) is an example of a biomarker gene that meets all of these criteria. Mammaglobin is a breast-specific transcript that has been previously identified as a blood biomarker for breast cancer detection[30]. In healthy breast tissue, mammaglobin is expressed at low levels, but cell transformation leads to a large increase in mammaglobin expression[31]. This is consistent with our data, which shows mammaglobin expression in plasma exclusively in patients with breast cancer for both our discovery and validation cohorts, with increased expression at later stages (Fig. 6a and Supplementary Fig. 12). Moreover, mammaglobin expression in plasma is specific to certain breast cancer subtypes in our cohort: expression is observed in the plasma of patients with HR+ breast cancer, but not in TNBC, consistent with expression in matched tumor tissue and previous studies[32].

Among the lung cancer DCB genes, many are associated with surfactant proteins (e.g., SFTA3, SFTPA2, SLC34A2, and BPIFA1), which function to lower surface tension at the air/liquid interface in alveolar cells. Interestingly, we observed that these lung markers are subtype-specific: they are detected in both plasma and matched tumor tissue samples from patients with lung adenocarcinoma (a type of tumor that originates from glandular epithelial cells in the smaller airways of the lung), but not in samples from patients with squamous cell carcinoma. The presence of these markers in the blood may benefit from the secretory nature of these cell types and their proximity to the blood/air interface, increasing the likelihood that these transcripts are released into the circulation during cell death or cell signaling. Surfactant-coding mRNAs have been previously identified as blood-borne biomarkers for subtype-specific detection of lung cancer[33], and there is substantial evidence in the literature that they are differentially expressed in lung adenocarcinoma tissue[34–36], suggesting an opportunity to leverage the vast literature around tumor tissue expression to identify additional subtype-specific cfRNA biomarkers.

We have also identified other types of DCBs that are not specific to a single tissue or cancer type. These comprise half of the DCBs identified in our study and are found in both breast and lung cancer patients within our cohort. AGR2 and TFF1, for example, are both estrogen-responsive gene elements that have been associated with a variety of different cancers. TFF1 has been reported as a blood-borne biomarker for breast cancer[37], while AGR2 has been previously reported as a prognostic marker for both breast cancer and lung adenocarcinoma[38,39]. Further studies of circulating RNA in different cancers, as well as cancers identified at earlier stages, will be needed to determine the value of these markers for cancer screening. On their own, they may represent binary markers of cancer status, while the presence of tissue-specific markers in the same sample will help predict tumor TOO.

Our results suggest that tumor-derived signals are amplified in cfRNA due to increased expression of these markers in tumor tissue. We introduce the notion of "tumor content" as a measure that accounts for both tumor shedding rate and tumor tissue expression that is more predictive of DCB detection in plasma than either metric alone. The ability to measure the tumor content of DCB genes through the analysis of cfRNA led to the amplification of cancer-specific signal by orders of magnitude compared to tumor fraction alone (Fig. 5). Moreover, the lack of appreciable levels of DCB genes in non-cancer plasma means that this amplification of tumor-specific signal does not lead to a concomitant increase in noise, enabling high specificity. In some cases, this increase in signal-to-noise may enable detection for patients with cancer with low tumor fraction that might otherwise be missed by cfDNA-based detection approaches.

Future studies of cfRNA and cfDNA from cancer patients with low tumor shedding but high tissue expression of DCB genes will be needed to directly address the potential increase in sensitivity afforded by a multianalyte test. Although we did not directly test complementarity with cfDNA-based detection on a sample-by-sample basis, our study identified cfRNA biomarkers that are specific to HR + breast cancer and lung adenocarcinoma, for which cfDNA has lower sensitivity compared to other breast and lung cancer subtypes[40,41]. Ultimately, it may be possible to supplement detection by cfDNA-based approaches through targeted enrichment of DCB transcripts.

There are several questions that must be addressed before the value of cfRNA for cancer screening in a clinical setting can be fully assessed. First, the sensitivity of cfRNA for cancer detection in early-stage cohorts will need to be established. This should be approached using a targeted assay, such as hybrid capture or

amplicon sequencing, which will allow more sensitive quantification of DCB genes that may have been missed in our cohort due to the low conversion efficiency of a whole-transcriptome approach. Future efforts should also be aimed at identifying additional DCB genes to further increase cancer detection sensitivity. Biomarkers can be identified empirically based on expression in TCGA tumor tissue for target cancers and filtered to remove biomarkers that are detected in the non-cancer transcriptome. Feature selection methods may also be expanded to include cancer-specific isoforms, fusions, and variants. Finally, assessment of cfRNA in larger non-cancer cohorts, and in patients with other breast-related or lung-related diseases, will be needed to establish the specificity of these markers in a screening population.

We now have a comprehensive picture of the types of plasma cfRNA in non-cancer individuals and have identified a class of cancer-specific cfRNA biomarkers in low-noise regions of the cell-free transcriptome. A subset of these cancer biomarkers are also subtype-specific, providing a potential strategy for both cancer detection and tissue-of-origin prediction. Beyond the individual biomarkers identified in this study for breast and lung cancer, we have created a workflow and computational framework for the identification of cfRNA biomarkers that is applicable to a variety of cancers. Ultimately, we hope cfRNA can enable the early detection of a variety of cancers to reduce cancer mortality.

## Methods

**Study design and sample selection criteria.** The protocol was reviewed and approved by the Institutional Review Board (IRB) or Independent Ethics Committee (IEC) for each of the 142 participating sites (the full list can be found at http://clinicaltrials.gov/ct2/show/NCT02889978). IRB Approval Letters, IRB Rosters, and Informed Consent Forms for each site are available in the Trial Master File and are available upon request. Informed written consent was obtained for each participant prior to sample collection. The Informed Consent Form contains the following statement: "The results of the study may be published in scientific journals and presented at medical meetings. The study doctor, study staff, and sponsor may make data, results, or biological samples from the study available in publicly accessible databases or provide them to other researchers for use in other research projects. If the data, results, or biological samples from this study are made public or provided to other researchers, information that directly identifies you will not be used."

The primary outcome of this study was to collect and study clinically-annotated biospecimens, specifically peripheral blood, and contemporary tumor tissue when available, to characterize cfRNA profiles from deep sequencing and to estimate the population heterogeneity in two arms of the study (cancer vs. non-cancer). For the discovery cohort, we selected a subset of stage III breast and lung cancer samples from the CCGA study (NCT02889978). Stage III samples were selected to maximize signal in the blood while avoiding confounding signal from potential secondary metastases. We required that the selected patients had at least 2 tubes of unprocessed grade 1–2 plasma (no hemolysis), with 6–8 mL of plasma per patient. We further required that selected patients had matched cfDNA sequencing data from previous GRAIL studies. We also selected an equal number of non-cancer samples matched for age, sex, and ethnicity to the cancer samples. Based on these criteria, a total of 172 patients were selected: 79 were diagnosed with stage III breast ($n = 47$) or lung ($n = 32$) cancer at the time of blood draw, plus 93 age-matched individuals without cancer. A subset of the patients with cancer also had matched breast ($n = 40$) and lung ($n = 12$) tumor tissue. For the validation cohort, we obtained 38 breast and 18 lung cancer samples from Discovery Life Sciences, and 38 age-matched non-cancer samples were included as controls of expression in patients without cancer.

**Sample collection.** Whole blood was collected in Streck Cell-free DNA BCT® tubes, which were shipped and stored at ambient temperature prior to plasma separation. Whole blood was centrifuged at $1600 \times g$ for 10 min at 4 °C to separate plasma. The plasma layer was transferred to a separate tube and centrifuged at $15,000 \times g$ for 12 min at 4 °C to further remove cellular contaminants. Double-spun plasma was stored at −80 °C and thawed at room temperature before extraction to avoid the formation of cryoprecipitates.

**Sample processing.** Cell-free nucleic acids were extracted from 8 mL of frozen plasma using the circulating miRNA protocol from the QIAamp Circulating Nucleic Acids kit (Qiagen, 55114). The reagent volumes were doubled to account for the increase in plasma volume compared to the manufacturer's recommended volume. When less than 8 mL of plasma was available, the volume was

supplemented with Dulbecco's phosphate-buffered saline (PBS). The extracted material was DNase treated using the RNase-free DNase Set (Qiagen, 79254) according to the manufacturer's instructions and quantified using the High Sensitivity RNA Fragment Analyzer kit (Agilent, DNF-472).

The whole-transcriptome assay was performed as follows. For each sample, 0.12 pg of External RNA Controls Consortium (ERCC) RNA reference standard was spiked into purified cfRNA as an in-line control for library preparation. Reverse transcription and adapter ligation were performed using the TruSeq RNA Exome kit (Illumina, 20020189) with the following modifications. We used custom adapters containing 8 base pair (bp) non-random UMIs. We also limited post-ligation amplification to 6 cycles of PCR and carried the entire yield into the depletion step. Libraries were depleted of abundant sequences using the AnyDeplete kit for Human rRNA and Mitochondrial RNA (Tecan, 9132), which was supplemented with depletion probes targeting HB subunits (*HBA1*, *HBA2*, *HBB*, *HBD*), components of the major histocompatibility complex (*B2M*, *CD74*), and the RNA component of the signal recognition particle (*RN7SL*). For targeted enrichment, the reverse transcription and adapter ligation steps were performed using the same protocol employed for the whole-transcriptome assay. Fifteen cycles of post-ligation PCR were performed, and up to 3.5 μg of material were used as input for targeted enrichment using the TruSeq RNA Enrichment protocol and a custom double-stranded probe panel manufactured by Twist Bioscience (South San Francisco, CA). Whole-transcriptome and targeted RNA-seq libraries were sequenced on a HiSeqX flowcell to a depth of 750 million and 100 million paired-end reads per sample, respectively.

**Sequencing data processing.** Raw reads were aligned to GENCODE v19 primary assembly with all transcripts using STAR version 2.5.3a. Duplicate sequence reads were detected and removed based on genomic alignment position and non-random UMI sequences. A majority of paired-end reads had UMI sequences exactly matching expected sequences. A subset of reads contained errors in the UMI sequence, and a heuristic error correction was applied. If the UMI was within a Hamming distance of 1 from an expected UMI, it was assigned to that UMI sequence. In cases where the Hamming distance exceeded 1, or multiple known sequences were within a Hamming distance of 1, the read with the UMI error was discarded. Sets of reads sharing alignment position and corrected UMIs were error corrected via multiple sequence alignment of member reads and a single consensus sequence/alignment was generated. Read alignments were compared to annotated transcripts in gencode v19.

All downstream analyses relied on the use of "strict RNA reads," defined as read pairs where at least 1 read overlapped an exon-exon junction (Supplementary Fig. 8). These reads are unique to RNA and help to filter DNA-derived background. Gene expression was normalized in RPM (reads per million mapped reads):

$$\text{RPM} = \frac{\text{Number of collapsed strict reads mapped to a gene} \times 10^6}{\text{Total number of collapsed strict mapped reads from library}} \quad (1)$$

Sequenced samples were screened and those exhibiting low quality-control metrics were excluded from subsequent analysis. One assay metric and 3 pipeline metrics were chosen as "red flags" and were used to exclude samples with poor metrics. The assay metric measured whether samples had sufficient material for sequencing (1.6 nM), and the pipeline metrics were sequencing depth, RNA purity, and cross-sample contamination.

## Statistical analysis

*Tissue deconvolution.* We used tissue deconvolution to estimate the relative contribution of each tissue type for cfRNA samples. Tissue deconvolution was performed by solving the following function:

$$y = X\beta + \epsilon \quad (2)$$

$$y \in \mathbb{R}^n, X \in \mathbb{R}^{n \times p}, \beta \in \mathbb{R}^p, \epsilon \in \mathbb{R}^n \quad (3)$$

$$min_\beta \left( \| X\beta - y \|^2 \right), s.t. \begin{cases} \sum_i \beta_i \leq 1 \\ \beta_i \geq 0, \forall_i \end{cases} \quad (4)$$

where $y$ is the observed expression data for $n$ signature genes in a cfRNA sample, $X$ denotes a median gene expression of the signature matrix for $p$ tissue types, $\beta$ is the fractional contribution of the tissue type toward the observed cfRNA sample, $\epsilon$ is the normally distributed error, and $i$ is the index of different tissue types. Because there might be other tissue types that were not included in $X$, we allow the summation of $\beta_i$ to be less than 1 and define the unexplained fraction $(1 - \sum \beta_i)$ as the remainder. To obtain the relative contributions of the tissue types, these equations were solved by quadratic programming to minimize the least-square error subjected to the above constraints.

To identify the signature genes for different tissues, we downloaded the gene expression matrix (htseq-count level) and sample annotations from the GTEx Portal website (www.gtexportal.org/home/datasets, GTEx analysis V4). We first normalized the gene expression in RPM and calculated the median RPM among all samples within each tissue type. Tissue specificity score (TSS) was calculated for

each gene. The TSS for gene $j$ is defined as

$$TSS_j = \max_i \left( x_{ij} / \sum x_{ij} \right) \quad (5)$$

$$i_j = \arg \max_i \left( x_{ij} / \sum x_{ij} \right) \quad (6)$$

where $x_{ij}$ is the median gene expression (in RPM) of gene $j$ in tissue $i$. When gene $j$ is tissue-specific, $i_j$ is the index of the tissue type for which gene $j$ is specific. For each tissue type, we selected the top 20 genes with the highest TSS as signature genes for tissue deconvolution.

Strict counts for cfRNA samples were compiled into gene expression matrices and normalized to RPM. We implemented quadratic programming using existing R programs and packages (quadprog, v1.5-7). Analysis and plots were generated using the R programming language (v3.6.2). As a consistency check, we also performed tissue deconvolution against the tissue RNA-seq samples using the same method as described above.

*Dark channel feature selection.* The dark channel genes were identified by 2 criteria: (1) The median expression (in RPM) of this gene in the non-cancer group is 0, and (2) the standard deviation of this gene is less than 0.1 RPM. The DCB for each cancer type were identified using 3 criteria: (1) There are at least 2 samples in the specified cancer group for which the gene is detected, (2) the RPM for the gene in the second-highest sample is greater than 0.1, and (3) the gene is differentially expressed in the specified cancer group compared to the non-cancer group ($P < 0.02$ for lung cancer and $P < 0.2$ for breast cancer). The $P$ value of 2-group differential expression was calculated by the edgeR (v3.26.3) package.

Annotation of tissue-specific genes was performed. The tissue-specific gene files for lung and breast tissues were downloaded from the Human Protein Atlas website (www.proteinatlas.org, version 18.1) and divided into 3 categories: (1) *Tissue Enriched*: At least 4-fold higher mRNA levels in a particular tissue as compared to all other tissues, (2) *Group Enriched*: At least 4-fold higher mRNA levels in a group of 2 to 5 tissues, and (3) *Tissue Enhanced*: At least 4-fold higher mRNA levels in a particular tissue as compared to average levels in all tissues. All 3 categories were included in our definition of tissue-specific genes.

*HeteroDE.* HeteroDE is an R package we developed to identify biomarker genes from highly heterogeneous plasma cfRNA samples. It models the abundance of RNA transcripts in plasma using a negative binomial generalized linear model (NB-GLM):

$$K_{i,j} \sim NB\left( \mu_{i,j}, \alpha_i \right) \quad (7)$$

$$\log(\mu_{i,j}) = \gamma_i + x_{i,j}\beta_i \quad (8)$$

where $i$ denotes a gene and $j$ denotes a patient, $K_{i,j}$ is the read count for gene $i$ in the cfRNA of patient $j$, $\mu_{i,j}$ is the expected read count for gene $i$ in the cfRNA of patient $j$, $\alpha_i$ is the dispersion for gene $i$, $\gamma_i$ is the intercept of the NB-GLM, $x_{i,j}$ is the tumor content (defined as the log10 transformed product of the tumor fraction in the matched cfDNA multiplied by the gene expression in the matched tumor tissue), and $\beta_i$ is the coefficient for the tumor content. Tumor content was used as the covariate in the model to account for the influence from both the gene expression in the tumor tissue and the tumor shedding rate. The tumor content for the non-cancer group was set to zero assuming the amount of RNA shedding from healthy lung or breast tissue into the blood is negligible, which is supported by Fig. 2. The tumor fraction was estimated from cfDNA mutation data. The $P$ value of the NB-GLM was computed using the glm.nb function in the MASS package (v7.3-51.4) in R. The $P$ value cut-off was set to 0.05 for the biomarker candidates.

HeteroDE also removes 2 types of false positives prevalent in highly heterogeneous samples. For the first type of false positive, the gene expression follows a bimodal distribution in both control and cancer groups due to genetic heterogeneity. The bimodal distribution disobeys the assumption of negative binomial distribution of the GLM and leads to spuriously low $P$ values. For the second type of false positive, a single influential outlier inflates the slope and $P$ value of the GLM. To reduce the false discovery rate, heteroDE includes 2 additional modules after the GLM step:

(1) HeteroDE checks if the gene expression of the identified biomarker follows the bimodal distribution. RNA-seq data has typically been modeled by a Poisson or negative binomial function[42]. We observe that the bimodal genes normally have one group of samples with minimal expression and the other group of samples with relatively high expression. We use the Poisson distribution to model the group with minimal expression and negative binomial distribution to model the group with relatively high expression[43]. To simplify the computational complexity, we approximated the negative binomial distribution to a Normal distribution. For a study with $n$ samples in the non-cancer group, each gene's expression profile is defined as $x(x_1, ...., x_n)$ which is assumed to be a random sample of a random variable $x$ whose density function can be written as:

$$p(x_i|\theta) = \pi_1 \text{Poisson}(x_i|\lambda_1) + \pi_2 \mathcal{N}\left(x_i|\mu_2, \sigma_2\right) \quad (9)$$

where $\pi_1$ and $\pi_1$ is the proportion of samples in component 1 and 2, respectively, and $\theta$ is the set of all parameters. The parameter estimation was carried out via the expectation-maximization algorithm. A bimodal gene is identified when $\pi_1$ is >0.1 and <0.9. If an identified biomarker gene has bimodal distribution in both non-cancer and cancer groups, it is flagged as false positive.

(2) HeteroDE checks if a single outlier sample is influencing the $P$ value of the NB-GLM. The Cook's distance for each sample was calculated using the Cook's distance function in R. The NB-GLM is performed for a second time without the sample with the largest Cook's distance. If the resulting $P$ value is no longer significant, this identified biomarker was flagged as a false positive.

**Reporting summary**. Further information on research design is available in the Nature Research Reporting Summary linked to this article.

## Data availability

All sequencing data have been deposited at the European Genome-phenome Archive (EGA) which is hosted at the European Bioinformatics Institute and the Centre for Genomic Regulation and is available under restricted access under accession number EGAS00001004704. Data access can be obtained through a request to the GRAIL Data Access Committee [https://www.ebi.ac.uk/ega/dacs/EGAC00001001769]. Access to the data will be restricted to non-commercial entities. Tissue-specific gene files for lung and breast tissues were downloaded from the Human Protein Atlas website (www.proteinatlas.org, version 18.1), tissue deconvolution was performed using gene expression data downloaded from the GTEx Portal website (www.gtexportal.org/home/datasets, GTEx analysis V4), and TCGA tumor tissue expression data are publicly available through the TCGA portal [https://portal.gdc.cancer.gov]. The remaining data are available within the article, Supplementary Information, or available from the authors upon request.

## Code availability

All code and scripts are available at https://github.com/grailbio-publications/Larson_cfRNA_DarkChannelBiomarkers[44].

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

## Acknowledgements

The authors thank Megan Hall for editorial assistance and critical reading of the article, Rick Klausner for helpful discussions, and David Burkhardt, Roger Jiang, Ajinkya Kokate, Archana Shenoy, and Xiao Yang for their help with data analysis. This study was funded by GRAIL, Inc.

## Author contributions

M.H.L., W.P., H.J.K., V.D., A.M.A. and A.J. contributed to project conceptualization and study design. M.H.L., S.M.S., M.P. and Y.Z. contributed to assay development and processed samples, and prepared libraries for sequencing. M.H.L., W.P., H.J.K., R.E.M. and P.K. contributed to data analysis and data visualization. M.H.L., V.D., A.M.A. and A.J. provided project supervision. All authors contributed to writing and editing.

## Competing interests

M.H.L., R.E.M., S.M.S., M.P., Y.Z., V.D. and A.J. are employees of GRAIL, Inc. with equity in the company. W.P., H.J.K., P.K. and A.M.A. are former employees of GRAIL, Inc. with equity in the company.
