## [Peer Review File · Nature Communications]

Reviewer #1, expert in systems biology, informatics (Remarks to the Author):

This manuscript by Larson, et al, describes an approach to analyze and characterize cell-free transcriptome to identify biomarkers for cancer detection. The study is quite thorough, including and analysis of the technical factors that may impact analysis of cfRNA. There are also several innovative ideas related to analysis and interpretation of cfRNA. Overall, this paper makes an important contribution to the field.

The major weakness is lack of rigorous analysis of relationship between DCB detection and tumor stage. An attractive possibility is for cfRNA to serve as a biomarker of early disease, as described in the discussion. However, as expected, the authors find that detection of DCB genes is associated with tumor stage. The authors should comment on the utility of this approach and the identified biomarkers for detection of early stage disease. Alternatively, the authors should discuss how to optimize these studies for identification and assessment of early stage biomarkers.

The underlying data and analytical pipelines are critical to the major claims. The authors must provide pointers to code (e.g., on github) as well as any R packages developed for these studies (e.g., HeteroDE). Additionally, the raw data must be made available at time of publication. The current statement about data and code availability is insufficient.

Reviewer #2, expert in cfDNA (Remarks to the Author):

The authors are correct in that a comprehensive assessment of cfRNA in individuals with and without cancer has not been conducted.

They performed the first transcriptome-wide characterization of cfRNA in cancer (stage III breast [n=46], lung [n=30]) and non-cancer (n=89) participants from the Circulating Cell-free Genome Atlas (NCT02889978). Of 57,820 annotated genes, 39,564 (68%) were not detected in cfRNA from non-cancer individuals. Within these low-noise regions, they identified tissue- and cancer-specific genes, defined as "dark channel biomarker" (DCB) genes, that were recurrently detected in individuals with cancer. DCB levels in plasma were correlated with tumor shedding rate and RNA expression in matched tissue, suggesting that DCBs with high expression in tumor tissue could enhance cancer detection in patients with low levels of circulating tumor DNA.

Intriguingly, cfRNA provides a unique opportunity to detect cancer, predict the tumor tissue of origin, and determine the cancer subtype. This is excellent and new work.

I have the following comments:

1. What might the role of cfRNA be - why does it exist or is it a by product and purely correlative? As this paper is potentially first in class this should be discussed.
2. The findings do not square with much literature out there on individual genes and cfRNA levels - how do the authors propose to deal with this?
3. A primary innovation in this work is that they restricted analysis to genes that are never (or rarely) detected in the non-cancer group, and identified biologically-relevant transcripts that are detected specifically in cancer plasma. They call these genes "dark channel biomarkers." I am not clear why this work was restricted in this manner. Why do they call these DCBs?
4. Since this is new to the field, it would be helpful for readers to know 'how difficult this is to do' and limitations of techniques. Is it the same for RNA from cells/tissues ie. short half life type issues.
5. What about detection of non-coding RNAs in the plasma - how do the authors deal with this particular issue.
6. Might these markers be prognostic or predictive, or tell us more?

7. I note discussions centres around specific genes eg. mammoglobin-A. But the literature on these genes does not correlate with their findings showing importance. How do the authors propose to deal with this?

Reviewer #3, expert in cfDNA bioinformatics (Remarks to the Author):

In this manuscript, the authors present a study on cfRNA isolated from blood plasma. After evaluation of some pre-analytical parameters, the authors present the identification of so-called dark channel biomarkers genes that are recurrently detected in cancer patients and can be used for (early) diagnosis of lung or breast cancer. While the concept of the DCB genes is interesting, this study is not convincing, lacks some important analyses and information as well as some important validation steps. Moreover, some results are regularly overinterpreted in unproven statements. Below I have listed several concerns.

Line 22-23:

"suggesting that DCBs with high expression in tumor tissue could enhance cancer detection in patients with low levels of ctDNA.": this is too suggestive in this manuscript. No results were presented to prove this statement.

Line 42-44

The authors state that "cfRNA presents an opportunity to detect cancer in patients with low tumor shedding rates, as overexpression of tumor-specific transcripts could ..." However, previous studies have shown that overexpressed RNAs in the tumor are not per se highly abundant in blood (e.g. Zeka et al., PMID: 30518699).

Line 63:

"We developed a strategy to preserve, extract, and sequence low-abundance mRNAs...": I don't see experimental prove in this manuscript that this was achieved for low-abundant mRNAs.

Line 73-76

Should be part of introduction.

Line 76-82 + supplementary pre-analytical testing line 1-106

As far as I know, the method to test the impact of pre-analytics has not been described before. This method should first be validated (using control samples and different tube types) before it can be used to claim that Streck tubes can be used to analyze cfRNA. Especially because recent studies have demonstrated that these tubes are not compatible with cfRNA analysis (PMID: 30935089). More specifically, by only looking at summed gene counts, the possible effect of pre-analytics on individual genes cannot be estimated accurately and comprehensively. Importantly, the specific effect on cancer related signals (i.e. cfRNA coming from the tumor cells) should be tested.

Figure 1a:

Is this one representative example of cfRNA fragment profiles from cancer patients? While a ribosomal RNA depletion was performed (M&M) it is strange to observe that the major RNA type observed using sequencing is ribosomal RNA. Concerning the composition of the mRNA fraction: is this not a representation of the capturing probes present in the TruSeqRNA-library prep?

Line 115:

Isn't it logical that the cfRNA is enriched in exonic regions compared to cfDNA, = the difference between DNA and RNA? What is the added value of this analysis?

Supplementary data line 38-53:

These are analysis results and should not be described in the (supplemental) methods section. More importantly, these results demonstrate that Streck tubes are not suitable for cfRNA analysis (also line 101-106). The authors highlight themselves that these tubes result in highly variable cfRNA analysis results. The conclusion that "by focusing only on cfRNA biomarkers with a low background in non-cancer samples, the methodological approach is validated" is simply incorrect;

variability between cancer samples caused by pre-analytics is not taken into account.

Line 88-97 and line 409-413

The authors indicate that cfRNA was extracted using the QIAamp Circulating Nucleic Acids kit. According to the manufacturer's manual this method can be used for sample volumes of maximum 5 ml. However, the authors indicate starting volumes of "up to 8 ml". How was RNA extraction performed? Also, the way this section is formulated, it seems that a variable plasma volume input amount was used for different samples. This will obviously result in different yields in cfRNA across different samples, and if no spike-in RNAs were used during RNA isolation that allow to correct for volume input in the analysis, samples cannot be compared.

Figure 2:

are the observed differences significant? A statistical test should be performed.

Line 136-138

These conclusions cannot be made based on the described findings and shown results in figure 2.

Line 156

Typo: WFCD2 instead of WFDC

Line 163:

the authors suggest an enrichment of tissue specific genes among the DCB genes: is this enrichment statistically significant (e.g. Fisher exact test)?

Figure 5:

Also for several cases with a high tumor content there is no expression of DCB gene in the plasma. Therefore, the conclusions made in lines 222-226 are not really supported by the data in this figure.

Line 238:

What is the added value of the 4 extra genes identified through heteroDE analysis? Do these markers perform better/equal as those identified through DCB criteria?

Line 245:

The specificity of the markers in cancer subgroups is not validated using the validation cohort?

Line 267:

An important question that remains: for how many cancer samples can these markers be used for diagnosis (what is the accuracy, specificity, sensitivity of a cfRNA based diagnostic test? And how does this compare to cfDNA markers/content?

Line 415

Concentration of ERCC spikes should be indicated.

Figure 4:

the legend doesn't match the figure: a, b, c, d, e, f annotation seems to be wrong.

Line 389:

Only stage 3 cases were selected for the first part of the study. Will important markers that are more specific for lower (or higher) stage tumors not be missed?

Open data

Links to the data should be included in the manuscript.

RESPONSES TO REVIEWERS

REVIEWER #1: Expert in systems biology, informatics

Remarks to the Author

This manuscript by Larson, et al, describes an approach to analyze and characterize cell-free transcriptome to identify biomarkers for cancer detection. The study is quite thorough, including an analysis of the technical factors that may impact analysis of cfRNA. There are also several innovative ideas related to analysis and interpretation of cfRNA. Overall, this paper makes an important contribution to the field.

RESPONSE: We thank the reviewer for their positive comments, including their assessment that we provided a “thorough analysis of technical factors that may impact analysis of cfRNA.” We also appreciate the reviewer’s remark that our work represents “an important contribution to the field.”

The major weakness is lack of rigorous analysis of relationship between DCB detection and tumor stage. An attractive possibility is for cfRNA to serve as a biomarker of early disease, as described in the discussion. However, as expected, the authors find that detection of DCB genes is associated with tumor stage. The authors should comment on the utility of this approach and the identified biomarkers for detection of early stage disease. Alternatively, the authors should discuss how to optimize these studies for identification and assessment of early stage biomarkers.

RESPONSE: With respect to the relationship between DCB detection and tumor stage, we note that the goal of this study was to comprehensively characterize cfRNA in non-cancer patients and to identify cfRNA biomarkers using late stage (stage III) cancer patients where there is an expectation of high tumor signal in the blood without confounding signal from potential secondary metastases. We sought to build a biological understanding about the types of signals present in both cancer and non-cancer plasma and then use that to guide future experiments (by us and others) to test the utility of these biomarkers and identify additional biomarkers present at earlier stages of cancer and in other cancer types. Future assay developments that increase the ability to capture rare transcripts in the blood (e.g., through more targeted methods) will also facilitate increased sensitivity for early-stage disease, just as the cfDNA field has evolved to detect cancer at earlier stages through method development.

We concede that we could have done more to spell out next steps and have added the following text to the discussion:

“There are several questions that must be addressed before the value of cfRNA for cancer screening in a clinical setting can be fully assessed. First, the sensitivity of cfRNA for cancer detection in early-stage cancer cohorts will need to be established. This could be approached using a targeted assay, such as hybrid capture or amplicon sequencing, which will allow more sensitive quantification of DCB genes that may have been missed in our cohort due to the low conversion efficiency of a whole-transcriptome approach. Future efforts should also be aimed at identifying additional DCB genes to further increase cancer detection sensitivity. Biomarkers can be identified empirically based on expression in TCGA tumor tissue for target cancers and filtered to remove biomarkers that are detected in the non-cancer transcriptome. Feature selection methods may also be expanded to include cancer-specific isoforms, fusions, and variants.”

The underlying data and analytical pipelines are critical to the major claims. The authors must provide pointers to code (e.g., on github) as well as any R packages developed for these studies (e.g., HeteroDE). Additionally, the raw data must be made available at time of publication. The current statement about data and code availability is insufficient.

RESPONSE: All sequencing data have been deposited at the European Genome-phenome Archive (EGA) which is hosted at the European Bioinformatics Institute and the Centre for Genomic Regulation, and will be publicly available under accession number EGAS00001004704. All code and scripts will be available at https://github.com/grailbio-publications/Larson_cfRNA_DarkChannelBiomarkers.

This link is included in the Data and Material Availability section in the main text of the manuscript.

REVIEWER #2: Expert in cfDNA

Remarks to the Author

The authors are correct in that a comprehensive assessment of cfRNA in individuals with and without cancer has not been conducted.

They performed the first transcriptome-wide characterization of cfRNA in cancer (stage III breast [n=46], lung [n=30]) and non-cancer (n=89) participants from the Circulating Cell-free Genome Atlas (NCT02889978). Of 57,820 annotated genes, 39,564 (68%) were not detected in cfRNA from non-cancer individuals. Within these low-noise regions, they identified tissue- and cancer-specific genes, defined as “dark channel biomarker” (DCB) genes, that were recurrently detected in individuals with cancer. DCB levels in plasma were correlated with tumor shedding rate and RNA expression in matched tissue, suggesting that DCBs with high expression in tumor tissue could enhance cancer detection in patients with low levels of circulating tumor DNA.

Intriguingly, cfRNA provides a unique opportunity to detect cancer, predict the tumor tissue of origin, and determine the cancer subtype. This is excellent and new work.

RESPONSE: We thank the reviewer for their positive comments, and agree that a comprehensive assessment of cfRNA in individuals with and without cancer has not been conducted. This was our primary motivation for this study, and we appreciate that the reviewer found our work to be both “excellent” and novel.

I have the following comments:

1. What might the role of cfRNA be - why does it exist or is it a by product and purely correlative? As this paper is potentially first in class this should be discussed.

RESPONSE: Although we too are curious about the origins of cfRNA in circulation, we do not have data to distinguish between different hypotheses. By design, our approach to cfRNA isolation is agnostic to the origin of the extracellular material and cannot distinguish between RNA released by living cells or during cell death (either necrosis or apoptosis).

We have added the following text in the discussion to address this point:

“By evaluating different preanalytical conditions and optimizing our extraction protocol, we were able to define a set of conditions for which cfRNA is preserved in plasma after blood collection. **By design, our approach to cfRNA extraction is agnostic to the origin of the extracellular material, and seeks to isolate all available RNA from the cell-free fraction. This precludes speculation about the potential function of these transcripts in circulation, but allows a comprehensive characterization of cfRNA.**”

2. The findings do not square with much literature out there on individual genes and cfRNA levels - how do the authors propose to deal with this?

RESPONSE: Studies that look at individual circulating mRNAs have also shown an increase in tumor-derived RNA in the blood¹⁻⁵. The limitation of these studies is that they focused on a limited number of

transcripts, as opposed to the whole-transcriptome approach detailed here. For circulating transcripts described in previous studies, we point out the consistency with our data (e.g., “Mammaglobin is a breast-specific transcript that has been previously identified as a blood biomarker for breast cancer detection³⁰”). To address the reviewer’s comment, we have added an additional statement to point out consistency with previously identified lung-cancer biomarkers:

“Surfactant-coding mRNAs have been previously identified as blood-borne biomarkers for subtype-specific detection of lung cancer³³, and there is substantial evidence in the literature that they are differentially expressed in lung adenocarcinoma tissue^{34–36}, suggesting an opportunity to leverage the vast literature around tumor tissue expression to identify additional subtype-specific cfRNA biomarkers.”

As the reviewer notes, this is a “first-in-class” study for cfRNA, so there is a paucity of data regarding the extracellular levels for the some of the biomarker transcripts identified in this study.

3. *A primary innovation in this work is that they restricted analysis to genes that are never (or rarely) detected in the non-cancer group, and identified biologically-relevant transcripts that are detected specifically in cancer plasma. They call these genes “dark channel biomarkers.” I am not clear why this work was restricted in this manner. Why do they call these DCBs?*

RESPONSE: The name “dark channel biomarkers” (DCBs) comes from the fact that these genes are found in regions of the genome (i.e., “channels”) that are largely unexpressed (i.e., “dark”) in non-cancer samples. We restricted our analysis to these genes to ensure high cancer specificity of the identified biomarkers. We found that the expression of a given transcript can range over orders of magnitude due to inherent biological differences and regardless of cancer status/stage (see examples below for 2 abundant circulating transcripts: PIK3CA and PTEN).

It is therefore difficult to set a threshold for fold changes for these abundant transcripts to ensure high specificity when trying to distinguish cancer from non-cancer. This constraint could be relaxed to increase the number of cfRNA cancer biomarkers and potentially the sensitivity of cancer-screening approach. However, it would come at the cost of decreased specificity.

We have added the following sentence to the discussion to provide more clarity on this point: “We call these genes “dark channel biomarkers.” These DCBs address a major challenge: identification of reliable RNA biomarkers for cancer detection while avoiding a large number of false positives. **By focusing**

our analysis on regions that are free of background signals from healthy cells, we can ensure high cancer specificity of the identified biomarkers. This approach effectively reduces the likelihood of identifying false positive cfRNA signal in plasma (e.g., due to technical variations or batch effects) that may arise from analyses focused on fold-changes in expression between cases and controls.”

4. *Since this is new to the field, it would be helpful for readers to know ‘how difficult this is to do’ and limitations of techniques. Is it the same for RNA from cells/tissues ie. Short half life type issues.*

RESPONSE: We look forward to publication of this work, as we attempted to describe our process with sufficient detail for others to replicate our results. The methods described here are well within the capacity of a standard molecular biology lab and do not require any special equipment. Perhaps the most challenging aspect of the work is the patient recruitment and high sequencing costs, which can be prohibitive for smaller labs. We therefore felt it was vital to share the sequencing data associated with this study. All sequencing data have been deposited at the European Genome-phenome Archive (EGA) which is hosted at the European Bioinformatics Institute and the Centre for Genomic Regulation, and will be publicly available under accession number EGAS00001004704. All code and scripts will be available at https://github.com/grailbio-publications/Larson_cfRNA_DarkChannelBiomarkers. This link is included in the Data and Material Availability section in the main text of the manuscript.

5. *What about detection of non-coding RNAs in the plasma - how do the authors deal with this particular issue.*

RESPONSE: We do include non-coding RNAs in our analysis, but they do not appear as statistically significant cancer biomarkers in our cohort.

6. *Might these markers be prognostic or predictive, or tell us more?*

RESPONSE: The identified DCBs are subtype-specific, and different subtypes of breast and lung cancer have been associated with different patient outcomes. As a result, subtype-specific DCBs are likely to be prognostic. Part of the CCGA study involves patient follow-up at regular intervals after blood draw. Future studies will allow us to better assess the prognostic power of the identified biomarkers.

7. *I note discussions centres around specific genes eg. Mammoglobin-A. But the literature on these genes does not correlate with their findings showing importance. How do the authors propose to deal with this?*

RESPONSE: There are several reports on plasma RNA that show specific detection of mammaglobin-A in breast cancer⁶⁻⁸. We have also added a comment about the concordance with circulating surfactant-related mRNAs in lung cancer patients (see our response to comment 2 above). We were gratified to see this concordance, but wondered why it had not translated to clinical practice. We believe it is the lack of highly specific cell-free mRNA biomarkers that has limited the translation of cfRNA into clinical practice for cancer screening.

REVIEWER #3: Expert in cfDNA bioinformatics

Remarks to the Author

In this manuscript, the authors present a study on cfRNA isolated from blood plasma. After evaluation of some pre-analytical parameters, the authors present the identification of so-called dark channel biomarkers genes that are recurrently detected in cancer patients and can be used for (early) diagnosis of lung or breast cancer.

RESPONSE: The goal of this study was to comprehensively characterize cfRNA in non-cancer patients and to identify cfRNA biomarkers using late-stage (stage III) cancer patients where there is an expectation of high tumor signal in the blood without confounding signal from potential secondary metastases. Future studies will seek to extend these results to early-stage cancers.

While the concept of the DCB genes is interesting, this study is not convincing, lacks some important analyses and information as well as some important validation steps. Moreover, some results are regularly overinterpreted in unproven statements. Below I have listed several concerns.

Line 22-23:

“suggesting that DCBs with high expression in tumor tissue could enhance cancer detection in patients with low levels of ctDNA.”: this is too suggestive in this manuscript. No results were presented to prove this statement.

RESPONSE: We present several examples of samples with low tumor fraction but high tumor tissue expression of *SCGB2A2*, which showed detectable levels of *SCGB2A2* in plasma: see samples BrCa9 (0.5% tumor fraction), BrCa27 (0.1% tumor fraction), and BrCa40 (0.1% tumor fraction) in Fig. 5b. We also show that tumor content was more significantly associated with DCB detection in plasma compared to either tumor fraction or tissue expression alone for the detectability of *FABP7* (Mann-Whitney U test $P=0.003$ vs 0.02 and 0.01, respectively) and *SCGB2A2* ($P=0.0001$ vs 0.01 and 0.001, respectively) in Fig. 5. Together, these 2 observations support the hypothesis that high expression in tumor tissue could enhance detection of DCBs in circulation. However, we agree that future studies using early-stage cancer patients with matched tissue and for which tumor fraction was determined to be empirically low using cfDNA will be needed to definitively prove/disprove this hypothesis.

Line 42-44

The authors state that “cfRNA presents an opportunity to detect cancer in patients with low tumor shedding rates, as overexpression of tumor-specific transcripts could ...” However, previous studies have shown that overexpressed RNAs in the tumor are not per se highly abundant in blood (e.g. Zeka et al., PMID:20518699).

RESPONSE: The referenced study (Zeka et al.) focused on microRNAs, which requires a different library preparation strategy (ligation of universal adapters versus the random hexamer priming strategy used here) and which may very well have different stability in the blood based on previous literature⁹. This paper also looks at serum, which is different from plasma due to the clotting step prior to processing. Moreover, this paper does not detail the collection tube used for blood collection, so it's not possible to directly compare with our study.

Studies that look at individual circulating mRNAs have also shown an increase in tumor-derived RNA in the blood¹⁻⁵. The referenced statement in the introduction (line 42-44) only serves to summarize previous literature for the reader and is not a conclusion reached by our work.

Line 63:

“We developed a strategy to preserve, extract, and sequence low-abundance mRNAs...”: I don’t see experimental prove in this manuscript that this was achieved for low-abundant mRNAs.

RESPONSE: The RNAs detected in our assay were present at low copy numbers (10’s of molecules per 20 mL of whole blood, see Figure 4a-b). Given the measured conversion efficiency of our assay ($4 \pm 3\%$, Fig. S8), this corresponds to 100’s of molecules per 8 mL blood draw. The tissue expression of these same biomarkers (Fig 4c-f), shows counts on the order of 10^3 - 10^5 for a single tissue sample. Compared to the counts seen in tissue or other cellular samples, the number of transcripts seen in circulation are relatively low. However, we agree with the reviewers point that this is a somewhat vague modifier, and have changed “low-abundance” to “extracellular” in the statement as follows:

“We developed a strategy to preserve, extract, and sequence **extracellular** mRNAs from patient plasma...”

Line 73-76

Should be part of introduction.

RESPONSE: We agree that this statement does not belong in the results section, and have removed it from the manuscript.

Line 76-82 + supplementary pre-analytical testing line 1-106

As far as I know, the method to test the impact of pre-analytics has not been described before. This method should first be validated (using control samples and different tube types) before it can be used to claim that Streck tubes can be used to analyze cfRNA. Especially because recent studies have demonstrated that these tubes are not compatible with cfRNA analysis (PMID: 30935089). More specifically, by only looking at summed gene counts, the possible effect of pre-analytics on individual genes cannot be estimated accurately and comprehensively. Importantly, the specific effect on cancer related signals (i.e. cfRNA coming from the tumor cells) should be tested.

RESPONSE: The referenced paper (by Sorber et al.) used Qiagen’s miRNeasy kit for RNA extraction, which uses a trizol lysis step followed by phenol-chloroform extraction. Consistent with the cited work, we also have internal results that suggest that trizol lysis followed by phenol-chloroform extraction is not compatible with Streck tubes, as it leads to significant clogging of the silica membrane and negligible cfRNA yields. Column-based approaches that use chaotropic salts (like the QIAamp Circulating Nucleic Acid kit used in our study) yield the same fragment profile for EDTA, Streck DNA, and Streck RNA blood collection tubes. This can be seen in the fragment length trace below.

Another significant difference between our work and the previous study from Sorber et al. was the amount of plasma used for extraction. Sorber and colleagues used only 200 μ L of plasma for extraction, whereas we used 8 mL, which is a 40-fold increase in volume. The minimal amount of plasma used in the previous work would have also substantially reduced the yield of cfRNA, making it difficult to quantify. When 8 mL of plasma is used as input into extraction, the recovered yields are more easily quantified and indicate an increase in cfRNA extracted from Streck blood collection tubes. Based on our observations of cfRNA yield

and fragment length distribution across tube types, we determined that Streck tubes would be more compatible with our chosen extraction method.

Comparison of cfRNA fragment length profiles and yields across different tube types. Blood was collected for 3 individual donors. For each donor, blood was collected in 3 different blood collection tubes: EDTA, Streck DNA, and Streck RNA. The plasma was pooled across donors for each tube type and extracted with the QIAamp Circulating Nucleic Acid Kit followed by DNase treatment. **a** Extracted cfRNA was run on the Bioanalyzer High Sensitivity RNA Analysis kit (Agilent) to visualize the fragment length distribution. The traces indicate that the cfRNA fragment length distribution is conserved across tube types. **b** Quantified extraction yields show an increase in cfRNA yield for Streck blood collection tubes compared to EDTA blood collection tubes.

The reviewer also suggests that our method for analyzing the effects of preanalytical parameters should be validated using control samples collected in different tube types. To address this comment, we conducted a study comparing Streck DNA vs Streck RNA tubes to determine if Streck RNA tubes provide significantly more stabilization of cfRNA in plasma.

We collected 4 tubes of whole blood from each of 7 donors. Blood was drawn into 2 Streck cfRNA tubes and 2 Streck cfDNA tubes for each donor. Whole blood was processed to plasma, and the plasma pooled according to tube type (Streck cfRNA vs Streck cfDNA) across patients, creating 2 pooled plasma fractions. After pooling, the samples were aliquoted into 16 x 8 mL fractions (8 tubes Streck cfRNA plasma and 8 tubes Streck cfDNA plasma) and extracted in parallel using the QIAamp Circulating Nucleic Acid kit.

The cfRNA yield from the different tube types was indistinguishable for fragments >200 nt, but showed an increase in yield for fragments <200 nt for blood collected in Streck cfRNA tubes (Wilcoxon rank-sum $P < 0.01$). To determine the effect on the final libraries and expression profiles, we prepared RNA-seq libraries for each aliquot of extracted cfRNA using the protocol described in the main text. RNA-seq libraries were enriched using a custom panel of 508 cancer-related genes, and each library was sequenced to saturation (~200 M reads per sample).

A comparison of collapsed gene counts for a given tube type provided a measure of assay reproducibility. A pairwise comparison of representative technical replicates from each tube type is shown below.

A comparison of collapsed gene counts across the different tube types provided a measure of tube-to-tube differences. A pairwise comparison of representative tube type samples is shown below.

The following graphs show the pairwise Pearson correlation for all samples in this study.

Pairwise correlations show that samples collected in the same tube type were more closely correlated with each other than with samples drawn in a different tube type, suggesting that technical replicates were highly

reproducible but that blood collection tube type affected the representation of transcripts in the cfRNA fraction. A differential expression analysis using DESeq2 indicated that 862 genes (of 63,664 total genes from Gencode v19 comprehensive gene annotation) were differentially expressed according to blood collection tube type ($P < 0.01$).

In general, we find that red blood cell-related transcripts, such as hemoglobin genes (*HBA1*, *HBA2*, *HBB*) were more highly represented in samples collected in Streck cfRNA tubes compared to the samples collected in Streck cfDNA tubes, suggesting increased hemolysis in Streck cfRNA tubes (see highlighted genes in the table below that details the top 10 differentially expressed genes). The apparent increase of mitochondrial genes in Streck cfDNA tubes further supports this hypothesis, as red blood cells lack mitochondria, leading to a relative reduction in the normalized expression of mitochondrial genes in Streck cfRNA upon increased hemolysis.

Higher in Streck DNA tubes			Higher in Streck RNA tubes		
Gene	Base Mean	Log Fold Change	Gene	Base Mean	Log Fold Change
MT-ND4	481.9851	2.722392	HBA1	1382.3384	-2.866163
MT-RNR2	58633.3445	2.758584	CDH1	1686.6945	-3.141931
MT-ND5	441.8311	2.663842	HBA2	2711.1871	-2.733049
MT-ND1	504.4979	2.943488	TFRC	22308.6291	-2.375259
MPL	9608.395	1.563541	TERC	7921.1984	-2.426289
MT-ND2	432.134	2.821219	PIM1	36849.0478	-2.405195
HLA-B	791.7665	1.813875	HBB	7531.5347	-2.735123
MT-CYB	284.6496	2.68949	RNY4	240.906	-4.065336
MT-RNR1	13554.3624	2.615123	ICOSLG	445.3151	-2.29739
MT-CO3	256.3742	2.807092	BCL2L1	74572.1088	-1.974307

Motivated by these differences in gene expression, we created a metric to measure the cfRNA fraction in any given sample and detect blood cell lysis. As described in the supplemental text, a total of 44 genes were found to be overexpressed in white blood cells and 32 were found to be overexpressed in cfRNA. We compared expression for these genes across tube types and determined that the Streck cfDNA tube had increased coverage for cfRNA genes (plot **a** in figure below), but that the Streck cfRNA tube had increased coverage for blood cell genes (plot **b** in figure below), suggesting increased hemolysis in Streck cfRNA tubes.

Comparison of summed counts for gene sets across different tube types. Log of the sum of either **a** cfRNA-enriched genes or **b** blood cell-enriched genes as a function of blood collection tube type. Asterisks represent significance levels for differential expression between cancer subtypes based on P values from the Wilcoxon rank-sum method: *** $P < 0.001$.

These results both support the conclusions of the individual gene expression analysis (i.e., that samples drawn into Streck cfDNA tubes show less evidence of hemolysis, and consequently less background noise, compared to samples collected in Streck cfRNA tubes) and validate our approach to measuring the effect of different preanalytical parameters through the analysis of specific gene sets. The results also support our selection of Streck cfDNA tubes for the preservation and analysis of cfRNA.

We have added a section describing this experiment and the associated analysis into the Supplemental Results.

Figure 1a:

Is this one representative example of cfRNA fragment profiles from cancer patients?

RESPONSE: This figure does not show cfRNA fragment profiles from cancer patients. The goal of this figure was to show what cfRNA looks like in a non-cancer patient. We thank the reviewer for pointing out this lack of clarity and have updated the figure legend to make this more apparent to the reader:

“**Fig. 1 Analytical characterization of cell-free RNA. a** Fragment Analyzer (Agilent) trace of cfRNA fragment lengths **in a non-cancer sample** following deoxyribonuclease (DNase) digestion. Inset: Relative proportion of different RNA types found by whole-transcriptome sequencing in a representative **non-cancer sample prior to abundant transcript depletion.**”

While a ribosomal RNA depletion was performed (M&M) it is strange to observe that the major RNA type observed using sequencing is ribosomal RNA. Concerning the composition of the mRNA fraction: is this not a representation of the capturing probes present in the TruSeqRNA-library prep?

RESPONSE: We apologize that this figure was not more clear. The pie chart shows RNA content **before** ribosomal RNA depletion. As such, there are no capture probes involved in the processing of the samples used for this plot. We have updated the figure legend to be more explicit:

“Inset: Relative proportion of different RNA types found by whole-transcriptome sequencing in a representative **non-cancer** sample **prior to abundant transcript depletion.**”

Line 115:

Isn't it logical that the cfRNA is enriched in exonic regions compared to cfDNA, = the difference between DNA and RNA? What is the added value of this analysis?

RESPONSE: We agree with the reviewer that it is logical that exonic regions would be enriched in cfRNA compared to cfDNA. This was a sanity check and part of the quality control assessment to convince ourselves and the reader that we are looking at mature RNA transcripts in the cell-free fraction. We have added the following statement to make this more explicit:

“...cfRNA shows increased coverage in exonic regions of expressed genes (Fig. 1b), **as expected for mature mRNA**, and reduced coverage in unexpressed genes (Fig. 1c), reflecting contributions from cell types with different expression profiles.”

Supplementary data line 38-53:

These are analysis results and should not be described in the (supplemental) methods section. More importantly, these results demonstrate that Streck tubes are not suitable for cfRNA analysis (also line 101-106).

RESPONSE: The results referenced on lines 38-53 refer to a drop in cfRNA coverage upon multiple freeze/thaw cycles (2 or more). However, a single freeze/thaw cycle (the condition used for this study) did not lead to a statistically significant change in cfRNA counts. The constraints on the size of the manuscript stipulated by the journal do not allow for a full description of preanalytical results in the main text, but we have added the following statement in the main text:

“**Based on these studies, we determined that Streck cfDNA blood collection tubes preserve cfRNA in whole blood at temperatures ranging from 6-35°C for up to 48 hours prior to plasma separation. Moreover, we found that cfRNA is stable in frozen plasma for up to 12 months when collected in Streck tubes and stored at -80°C, and that a single freeze/thaw cycle does not significantly affect cfRNA yield, enabling** the use of banked plasma for the study of cfRNA.”

The authors highlight themselves that these tubes result in highly variable cfRNA analysis results. The conclusion that “by focusing only on cfRNA biomarkers with a low background in non-cancer samples, the methodological approach is validated” is simply incorrect; variability between cancer samples caused by pre-analytics is not taken into account.

RESPONSE: The statement about variability in lines 50-52 was not in reference to tube types but rather the effect of processing samples at different times and by different operators. Preanalytical variability is inherent to any clinical experiment, and our work stands out in our attempt both to address the effect of these different parameters and to control them in the collection of samples for our clinical study.

Nevertheless, we have removed this statement from the supplement for clarity and because it is not a primary conclusion of this work.

Line 88-97 and line 409-413

The authors indicate that cfRNA was extracted using the QIAamp Circulating Nucleic Acids kit. According to the manufacturer's manual this method can be used for sample volumes of maximum 5 ml. However, the

authors indicate starting volumes of “up to 8 ml”. How was RNA extraction performed? Also, the way this section is formulated, it seems that a variable plasma volume input amount was used for different samples. This will obviously result in different yields in cfRNA across different samples, and if no spike-in RNAs were used during RNA isolation that allow to correct for volume input in the analysis, samples cannot be compared.

RESPONSE: The QIAamp protocol used in our study was the “Purification of Circulating microRNA from Serum, Plasma, or Urine.” We doubled the reagent volumes specified by the protocol to account for the additional plasma volume used in our study. Each plasma sample was also normalized using PBS so that all samples had exactly 8 mL of plasma prior to extraction. The methods were updated to include the following statement:

“Cell-free nucleic acids were extracted from 8 mL of frozen plasma using the circulating miRNA protocol from the QIAamp Circulating Nucleic Acids kit (Qiagen, 55114). **The reagent volumes were doubled to account for the increase in plasma volume compared to manufacturer’s recommended volume. When less than 8 mL of plasma were available, the volume was supplemented with Dulbecco’s phosphate buffered saline (PBS).**”

The yield is presented as a concentration (nanograms per milliliter of plasma), which accounts for the original volume of plasma prior to normalization. As such, this allows comparison of cfRNA concentration across different patients (though the total mass in nanograms will certainly be dependent on the plasma volume).

Figure 2:

are the observed differences significant? A statistical test should be performed.

RESPONSE: (See our response to the next comment, which is related to this comment.)

Line 136-138

These conclusions cannot be made based on the described findings and shown results in figure 2.

RESPONSE: We thank the reviewer for bringing this to our attention. We conducted a Wilcoxon rank-sum test on the deconvoluted breast and lung fractions between the non-cancer, breast cancer, and lung cancer groups. In the tumor tissue, the breast fraction was significantly higher in breast tissue samples than in lung tissue samples ($P = 9.765e-07$). In the tumor tissue, the lung fraction was significantly higher in lung tissue samples than in breast tissue samples ($P = 0.0008722$). In plasma, the lung fraction was not significantly higher in lung cancer samples than in breast cancer samples ($P = 0.4925$) or non-cancer samples ($P = 0.1366$). In the plasma, the breast fraction was not significantly higher in breast cancer samples than in lung cancer samples ($P = 0.161$) or non-cancer samples ($P = 0.9914$).

We have added indications of significance in Fig. 2b and have amended the text to remove any suggestion that there is a statistically significant increase in tissue-specific transcripts in plasma from breast or lung cancer patients. The statement now simply confirms that tissue-specific transcripts were *observed* in cancer plasma, which serves to motivate the next results section where we seek to identify such transcripts. It now reads as follows:

“We also performed a tissue deconvolution analysis on RNA from the cancer group, using plasma and matched tumor tissue samples obtained for the CCGA study. The results of these analyses revealed larger

contributions from breast and lung tissue in RNA from tumor tissue samples (Fig. 2b), as expected. **Notably, lung and breast-derived transcripts were also observed in cfRNA from cancer plasma samples (Fig. 2c). The relative fraction of these tissue-specific transcripts observed in plasma is low compared to estimates in tumor tissue, as expected based on previous estimates of circulating tumor DNA in plasma⁶, but are indicative of tissue-specific transcripts in cancer plasma that can be differentiated from the high background of blood-cell derived transcripts. Overall, this suggests that there is a detectable population of tumor-derived RNA circulating in cancer plasma.**"

Line 156

Typo: WFCD2 instead of WFDC

RESPONSE: Thank you for catching this. We have fixed this typo in the manuscript.

Line 163:

the authors suggest an enrichment of tissue specific genes among the DCB genes: is this enrichment statistically significant (e.g. Fisher exact test)?

RESPONSE: Lung DCB genes are significantly enriched for lung-specific genes (Fisher's exact $P = 7.14 \times 10^{-13}$), and breast DCB genes are significantly enriched for breast-specific genes (Fisher's exact test $P = 2.69 \times 10^{-9}$).

We have amended the text to include this statement:

"By comparison, 50% of the lung DCB genes (6 of 12) are lung-specific, and 50% of the breast DCB genes (4 of 8) are breast-specific, **indicating significant enrichment of tissue-specific markers for both lung (Fisher's exact test $P = 7.1427686 \times 10^{-13}$) and breast DCBs (Fisher's exact test $P = 2.6924765 \times 10^{-9}$).**"

Figure 5:

Also for several cases with a high tumor content there is no expression of DCB gene in the plasma. Therefore, the conclusions made in lines 222-226 are not really supported by the data in this figure.

RESPONSE: The reviewer is correct that there are some high tumor content samples where there is no detection of the DCB gene in plasma. In some cases this is because that particular marker is not expressed in the matched tumor tissue sample (e.g., BrCa31 for *FABP7*, or BrCa26 for *SCGB2A2*). In other cases, this may arise due to varying levels of RNA degradation or assay efficiency from sample to sample. We have amended the results section to include the following text:

"There are also a number of samples that appear to have relatively high tumor content (>1%) for *SCGB2A2* and *FABP7*, but for which the genes are not detected in cfRNA, suggesting that other factors beyond tumor tissue expression may influence the detectability of a given biomarker in circulation."

Line 238:

What is the added value of the 4 extra genes identified through heteroDE analysis? Do these markers perform better/equal as those identified through DCB criteria?

RESPONSE: There are 4 additional genes identified using HeteroDE (*CRABP2*, *VGLL1*, *SERPINB5*, and *TFF1*). All show up as highly significant biomarkers for at least 1 cancer type (see Fig 6), highlighting the value of this method in identifying additional useful cfRNA biomarkers.

Line 245:

The specificity of the markers in cancer subgroups is not validated using the validation cohort?

RESPONSE: We validated the DCBs in a separate validation cohort using commercially-sourced samples (Discovery Life Sciences). Most of the cancer patients in this cohort do not have reliable subtype data available at the time of collection. As such, we were able to validate DCBs as specific markers of either breast or lung cancer, but we were not able to validate their subtype specificity (i.e., HR+ breast cancer vs TNBC, or lung adenocarcinoma vs squamous cell carcinoma) in this cohort. It is worth noting, however, that the subtype specificity of DCB genes is supported by expression of these markers in TCGA tumor tissue, which is a separate cohort outside of our study that further validates the subtype-specificity of these markers.

We have added the following statement in the validation section of the results to clarify this point for the reader:

“Most of the cancer patients in this cohort did not have reliable subtype data available at the time of collection, so this cohort could only be used to validate the cancer specificity of DCB genes and not their subtype specificity.”

Line 267:

An important question that remains: for how many cancer samples can these markers be used for diagnosis (what is the accuracy, specificity, sensitivity of a cfRNA based diagnostic test? And how does this compare to cfDNA markers/content?)

RESPONSE: Classification performance is out-of-scope for this study, as it was designed as a discovery study aimed at understanding the biology of cfRNA in non-cancer patients and identifying cfRNA biomarkers in late stage breast and lung cancer. This study was not powered to assess the clinical performance of cfRNA for cancer classification. We agree with the reviewer that this is an important area of inquiry for the field, and future studies with a larger and more diverse set of cancer samples spanning different stages will help assess the degree to which cfRNA can supplement cfDNA classification performance.

Line 415

Concentration of ERCC spikes should be indicated.

RESPONSE: The current text stipulates the mass of ERCC added per sample:

“For each sample, 0.12 pg of External RNA Controls Consortium (ERCC) RNA reference standard was spiked into purified cfRNA as an in-line control for library preparation.”

The mass of ERCC added is the most relevant parameter, as it determines the absolute number of ERCC molecules present in the final library.

Figure 4:

the legend doesn't match the figure: a, b, c, d, e, f annotation seems to be wrong.

RESPONSE: We thank the reviewer for bringing this to our attention. We have corrected the annotations in the figure image.

Line 389:

Only stage 3 cases were selected for the first part of the study. Will important markers that are more specific for lower (or higher) stage tumors not be missed?

RESPONSE: The goal of this study was to comprehensively characterize cfRNA in non-cancer patients and to identify cfRNA biomarkers in late-stage (stage III) cancer patients where there is an expectation of high tumor signal in the blood without confounding signal from potential secondary metastases. We sought to build a biological understanding about the types of signals present in both cancer and non-cancer plasma and then use that to guide future experiments (by us and others) to identify additional biomarkers present at early stages of cancer and in other cancer types. With this groundwork laid, we can begin to establish the clinical performance of cfRNA (both independently and in combination with other analytes) in future studies.

Open data

Links to the data should be included in the manuscript.

RESPONSE:

All sequencing data have been deposited at the European Genome-phenome Archive (EGA) which is hosted at the European Bioinformatics Institute and the Centre for Genomic Regulation, and will be publicly available under accession number EGAS00001004704. All code and scripts will be available at https://github.com/grailbio-publications/Larson_cfRNA_DarkChannelBiomarkers. This link is included in the Data and Material Availability section in the main text of the manuscript.

Reviewer #1 (Remarks to the Author):

This study makes an important contribution to the field and is likely to advance the understanding and use of cell free RNA in the research and clinical settings. The authors have addressed my concerns and I have no additional comments.

Reviewer #2 (Remarks to the Author):

Congratulations
Great paper
Justin Stebbing

Reviewer #3 (Remarks to the Author):

Review of rebuttal reviewer 3

In general the authors have answered the questions appropriately and have made the needed changes. However, a few comments remain.

As from comment on Line 267 and Line 389, in the manuscript the authors should clearly explain the goal of this study and that it doesn't reach towards the identification and validation of new biomarkers and comparison with cfDNA-based biomarkers.

Authors might also comment in the discussion on the fact that the detected DCBs might also be detectable in patients with other breast- or lung-related diseases than cancer, and that therefore further analyses are needed to check the specificity of these markers.

RESPONSES TO REVIEWERS

REVIEWER #3: Expert in cfDNA bioinformatics

Remarks to the Author

As from comment on Line 267 and Line 389, in the manuscript the authors should clearly explain the goal of this study and that it doesn't reach towards the identification and validation of new biomarkers and comparison with cfDNA-based biomarkers.

RESPONSE: Per the request of reviewer #3 and the editor, a statement regarding the goal and primary outcome of this study was added to the study design section of the manuscript:

“The primary outcome of this study was to collect and study clinically-annotated biospecimens, specifically peripheral blood and contemporary tumor tissue when available, to characterize cfRNA profiles from deep sequencing and to estimate the population heterogeneity in two arms of the study (cancer vs non-cancer).”

We have also softened the language in line 267 to read:

“In summary, these results **confirm the diagnostic potential of 15 of the 23 DCB genes in individuals with cancer** and validate the DCB discovery approach.”

Similarly, line 389 now reads:

“A subset of these cancer biomarkers are also subtype-specific, providing a **potential** strategy for both cancer detection and tissue-of-origin prediction.”

Authors might also comment in the discussion on the fact that the detected DCBs might also be detectable in patients with other breast- or lung-related diseases then cancer, and that therefore further analyses are needed to check the specificity of these markers.

RESPONSE: We have added a sentence in the Discussion to highlight the necessity to validate the specificity of DCBs across larger non-cancer populations, including those with breast or lung-related diseases:

“Finally, assessment of cfRNA in larger non-cancer cohorts, and in patients with other breast- or lung-related diseases, will be needed to establish the specificity of these markers in a screening population.”